# Hydrochemical Characteristics of Hot Springs in the Intersection of the Red River Fault Zone and the Xiaojiang Fault Zone, Southwest Tibet Plateau

Zirui Li [1], Xiaocheng Zhou [2,*], Qiulong Xu [3,*], Yucong Yan [2], Miao He [2], Jingchao Li [2], Jinyuan Dong [2], Jiao Tian [2], Fenghe Ding [1], Ying Li [2], Chongzhi Ma [1], Xiaotao Wang [1] and Zhixin Luo [2]

[1] Ningxia Earthquake Agency, Yinchuan 750001, China
[2] United Laboratory of High-Pressure Physics and Earthquake Science, Key Laboratory of Earthquake Prediction, Institute of Earthquake Forecasting, CEA, Beijing 100036, China
[3] Earthquake Agency of Xinjiang Uygur Autonomous Region, Urumqi 830011, China
[*] Correspondence: zhouxiaocheng188@163.com (X.Z.); ytftodx@163.com (Q.X.)

**Abstract:** The coupling relationship between regional seismic activity and the hydrogeochemical field provides an important theoretical basis for regional earthquake precursor exploration. The intersection area of the Red River fault zone (RRF) and the Xiaojiang fault zone (XJF) in southeast Yunnan province has become the focus area of earthquake monitoring and prediction because of its special tectonic position in China. There were 20 hot springs that were sampled and analyzed in the laboratory for major elements, including trace elements, silica, stable isotopes ($\delta^{18}$O and $\delta$D), and strontium isotopes, from the years 2015 to 2019. (1) The meteoric water is the main source of recharge for thermal springs in the study area, and recharged elevations ranged from 1.1 to 2 km; (2) the geothermometer method was used to estimate the region of thermal storage temperature, and its temperature ranged between 64.3 to 162.7 °C, whereas the circulation depth ranged from 1.1 to 7.2 km. Hydrochemical types were mainly controlled by aquifer lithology, in which sodium bicarbonate and sulphuric acid water gathered mainly in the RRF, while calcium bicarbonate water gathered mainly in the XJF. According to the silicon–enthalpy equation method, the temperature range and cold water mixing ratio were 97–268 °C and 61–97%, respectively; (3) the circulation depth of the RRF was deeper than that of the XJF, and it was mainly concentrated in the second segment and the fourth segment on the RRF. Most of the hot spring water was immature with a weak water–rock reaction; (4) the hot water intersections of RRF and XJF were obviously controlled by the fault and the cutting depth of granite; (5) the relationship discussed between geothermal anomaly and earthquake activity had a good correspondence with regional seismicity. The intensity of the reaction between underground hot water and the surrounding rock may lead to the change of pore pressure, and the weakening effect of groundwater on fracture may change accordingly, followed by the change in the adjustment of tectonic stress. Eventually, the difference in seismic activity was shown, implying that deep fluid has an important control action on the regional seismicity.

**Keywords:** hot spring; hydrogeochemistry; isotope; geothermal; Red River fault zone; Xiaojiang fault zone





## 1. Introduction

A hot spring is a window to reflect the changes in deep crust and the groundwater circulation system. In recent years, more and more attention has been paid to the stud of geochemical characteristics of active faults in seismology [1,2]. The studies of geothermal water chemistry can not only solve the source of geothermal water and chemical components, geothermal storage temperature, the mixing ratio of hot and cold water, geothermal water circulation depth, and water–rock reaction degree but also help to better understand

the fault movement characteristics, such as the mode of activity, activity intensity, and cutting depth of the fault. Discussion on the geothermal genesis mechanism and information on earthquake preparation [3–5] were reviewed previously.

The latest research progress shows that the deep source flow, rich in the asthenosphere and the low-velocity layer in the crust, is discharged on the surface through the faults or fissures in the lithosphere or the upper crust, thus carrying the signals of the upwelling of the asthenosphere and the evolution of the low-velocity layer in the crust [6,7]. When these fluids intrude into faults, they can increase pore pressure, reduce frictional force in fault zones, and promote fault movement to produce earthquakes [8–10]. Geophysical evidence suggests that the direct cause of seismic activity on the San Andreas fault near Parkfield and Cholame is high-pressure fluid from the upper mantle [11]. Xu [12] believed that the birth and occurrence of earthquakes were related to the high conductivity and low-velocity layers in the crust through the study of the 1976 Tangshan earthquake.

In this paper, the intersection area of the RRF and the XJF, in the Sichuan–Yunnan rhomboid block, was selected as the research domain to analyze the hydrogeochemical characteristics of 20 hot springs from the years 2015 to 2019, including the flow path inside the fault, the change of heat storage, the source of significant substances, the water–rock reaction, and the connection with earthquakes. The novelty of this paper was that the hydrochemical index confirms the existence of high conductivity and low velocity layers in the intersection area of the RRF and the XJF, and they may be related to seismic activity. It provides basic data for understanding the relationship between geothermal anomaly, regional seismic activity, and regional seismic hazard discrimination.

## 2. Study Area and Geological Settings

The study area in this paper includes the intersections of the RRF and the XJF, which are two large fault zones on the Southeastern margin of the Sichuan–Yunnan block. Note that the RRF in this study refers to the southern section of RRF, and the XJF refers to the southern section of the XJF, including the east branch and the west branch. Details of the spring groups studied were illustrated in Figure 1 and Table 1.

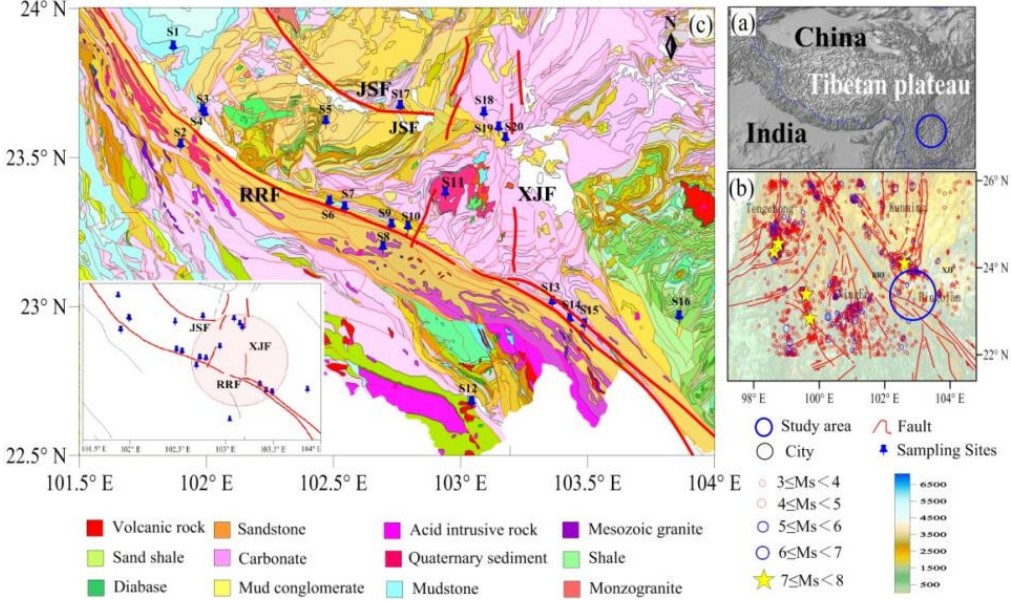

**Figure 1.** The plot of sampling site distribution. (**a**) Localization of the area of this study. (**b**) Topographic and earthquake distribution map in the intersection of the RRF and the XJF. RRF: Red River Fault Zone; XJF: Xiaojiang Fault Zone; (**c**) Geological map in the intersection of the RRF and the XJF.

**Table 1.** The location and lithology of hot spring in the intersection area of the RRF and the XJF.

| No. | Spring Name | Longitude (°) | Latitude (°) | Altitude (m) | Belongs to Fracture | Outcropped Lithology * |
|---|---|---|---|---|---|---|
| S1 | Lei Da | 101.8694 | 23.84618 | 445 | RRF | carbonate |
| S2 | Wa Na | 101.8983 | 23.51716 | 941 | | Triassic Ganbatang metamorphic sandstone and phyllite |
| S3 | Qi Xia | 101.9847 | 23.63402 | 413 | | sandstone, carbonate |
| S4 | Tian Cheng | 101.9937 | 23.6226 | 374 | | sandstone, carbonate |
| S5 | Ma Anshan | 102.4693 | 23.59537 | 1052 | | carbonate, limestone |
| S6 | Hu Jie | 102.4851 | 23.32551 | 414 | | Ailaoshan group schist, marble, mixed rock |
| S7 | Hong Yuan | 102.5441 | 23.30633 | 297 | | metamorphic rock |
| S8 | Po Duo | 102.6953 | 23.17386 | 1144 | | Ailaoshan group schist, marble, mixed rock |
| S9 | Nan Shan | 102.729 | 23.24808 | 514 | | sandstone, metamorphic rock, carbonate |
| S10 | Pai Shan | 102.7938 | 23.24111 | 286 | | sandstone, metamorphic rock, carbonate |
| S11 | Ge Jiu | 102.9396 | 23.35529 | 886 | | Jurassic granite |
| S12 | Meng La | 103.0441 | 22.65327 | 432 | DDFF | carbonate, limestone |
| S13 | Xiao Madian | 103.3615 | 22.99038 | 320 | RRF | Ailaoshan group schist, marble, mixed rock |
| S14 | Meng Qiao | 103.4294 | 22.93215 | 240 | | Ailaoshan group schist, marble, mixed rock |
| S15 | Teng Long | 103.4884 | 22.91374 | 136 | | Ailaoshan group schist, marble, mixed rock |
| S16 | Pin Bian | 103.8604 | 22.94037 | 274 | MZ-NXHF | sandstone, volcanic rock |
| S17 | Huang Longsi | 102.7625 | 23.64655 | 1334 | XJF | limestone |
| S18 | Da Tianshan | 103.0921 | 23.62381 | 1194 | | carbonate, limestone |
| S19 | Wen Shuitang | 103.1497 | 23.57432 | 1168 | | carbonate, limestone |
| S20 | Long Yuan | 103.1765 | 23.53867 | 1183 | | carbonate, limestone |

"*" is quoted from geological cloud 2.0 1/200,000 geological map of China; DDFF: Dien bianfu fault Zone; MZ-NXHF: Mengzi-Nanxihe fault Zone; RRF: Red River fault Zone; XJF: Xiaojiang fault Zone.

The RRF is the most complex active tectonic characteristic, the most frequent strong seismic activity zone, and one of the research hot springs in the Tibetan Plateau, as revealed recently. It is a major scale and complex tectonic evolution of the Cenozoic dextral strike–slip active fault zone [13,14] with a total length of more than 1000 km. Its maximum left-hand strike–slip occurs at 30–20 Ma [15] with a strike–slip distance of 500 km. The RRF is divided into three parts: North, Central, and South. The northern section is from Eryuan north to Juli, the middle section is from Juli to the southeast of Chunyuan, and the southern section is from Chunyuan south to Hekou. The strike–slip rates in the northern, middle, and southern sections are 3.6 mm·a$^{-1}$, 2.7 mm·a$^{-1}$, and 2.8 mm·a$^{-1}$, respectively [16]. Since the tertiary, the fault has taken place in the sinistral, dextral, and local rotation, and the activities of the north and south parts of it are unique to some extent. On the one hand, the northernmost segment is active, and there have been a lot of earthquakes with $M_S \geq 6.0$. On the other hand, the activities in the southern segment are quiet nowadays. One view is that the boundary effect of the RRF is weakened, and there will be no major earthquakes in the future. Another point of view is that the middle and southern segments of the RRF have a long recurrence cycle and may be the gestation period of large earthquakes, suggesting that there is still a risk of large earthquakes in the future [17].

The XJF, which is an important aspect of the north–south seismic belt of China, is the southeast boundary of the Sichuan–Yunnan rhomboid block. It is an active fault with strong neurogenic capacity and great destructibility. It is another important area for understanding the Geophysical dynamics of the southeastern margin of the Qinghai–Tibet Plateau. Since

1500, there have been more than 20 earthquakes above $M_S$ 6.0 in this fault zone [18,19], including the Yiliang $M_S$ 7.0 earthquake in 1500, the Dongchuan $Ms$ $7^{3/4}$ earthquake in 1733, the Huaning $M_S$ 7.0 earthquake in 1789, and the Songming $M_S$ 8.0 earthquake in 1833 [20,21]. These earthquakes have caused people's lives lost and property loss. For the near century, the southern end of the XJF has had a few historical seismic records. However, the recent slip rate obtained by GPS also represented a rate of 7–10 mm·$a^{-1}$ [22,23]. On space distribution, some scholars [24,25] held that the southern segment of the XJF did not extend southward to intersect with the RRF and ended near Shanhua Village, Jianshui County. However, newer tectonic models [26,27] believed that the southern segment of the XJF was cut southward. The RRF connected with the NE-trending Dian Bianfu fault forms the eastern boundary of the Qinghai–Tibet plateau. According to the mantle-derived helium release intensity along the boundary of the Sichuan–Yunnan Rapid uplift structure [28], it can be inferred that the deep cutting of the XJF is limited to the crust.

Owing to the strong tectonic movement in the vicinity of the RRF, the strata from Cambrian to Devonian appear angular unconformity contact. The strata are Precambrian, mainly act as teleprinters (750–970 Ma), and are outcropped in the Yangtze Craton along the Ashan red river fault zone [29], with granitic gneiss, migmatite, amphibole, and mica schist. These strata are covered with Paleozoic sedimentary rocks (conglomerate, quartzite, greenschist, black schist, sericite, carbonate, etc.), some of which are partially metamorphic in some places. In addition to sedimentary and metasomatic rocks, some magmatic rocks are likewise exposed.

The XJF has a complex stratigraphic structure, and the strata are exposed from the Middle Proterozoic to Cenozoic. The relatively old ones are phyllite and slate formed by metamorphism of the Middle Proterozoic Huangcaoling Formation and Meiding Formation, including quartzite, sandstone, stromatolite, and limestone of the Middle Proterozoic Heishan Formation and Dalong Kou Formation. It is to be noted that Sinian sandstone, conglomerate, tuff, and so on are close to the Middle Proterozoic. In local areas, the upper layer is composed of carbonate, whereas the lower layer is the composition of sandstone and conglomerate.

At present, there are three periods of magmatic activity in the study area, comprising the intestines and earlier acidic magmatic activity in the south side of the AiLaoShan metamorphic belt, which are mainly granite and rhyolite. There are three archetypal Yanshanian granite intrusions (Gejiu, Bozhushan, and Laojunshan) on the north segment of the RRF [30]. They are related to the Cenozoic potassium basaltic magmatism and Pingbian basaltic volcanic eruption of MaguanLaojun Mountain on the north side of the RRF [31].

## 3. Materials and Methods

### 3.1. Experimental Techniques

The groundwater samples were obtained from 20 springs, along the intersections of the RRF and the XJF, in March 2015, January 2016, February 2017, March 2018, and May 2019, respectively (Figure 1c). These water samples were analyzed by principal anions and cations, trace elements, hydrogen and oxygen isotopes, and $SiO_2$. Water samples were put into clean polyethylene bottles, which were thoroughly rinsed 2~3 times using the groundwater to be sampled. The concentrations of cations ($K^+$, $Na^+$, $Mg^{2+}$, and $Ca^{2+}$) and anions ($F^-$, $Cl^-$, $NO_3^-$, $SO_4^{2-}$) were measured with the Dionex ICS-900 ion chromatography and AS40 automatic sampler, with the reproducibility within ±2% in the Seismic Fluid Laboratory of the Institute of Earthquake Science, China Earthquake Administration. The $CO_3^{2-}$ and $HCO_3^-$ concentrations were measured by standard titration procedures with a ZDJ-100 potentiometer titrator (reproducibility within ±2%). Diluted hydrochloric acid, with alkalinity of 0.05 mol/L, was used in site titration. The trace elements analysis was performed by inductively coupled plasma mass spectrometry technique ICP-MS [32]. The quality of the chemical data was assessed by calculating ion

balances (ib). All ib were found below 5%; therefore, all samples were considered in the calculations of this work. The ib were calculated according to the Equation (1) [5].

$$ib[1\%] = \frac{\sum cations - \sum anions}{(\sum cations + \sum anions) \times 0.5} \times 100 \tag{1}$$

Water samples for the analysis of $\delta^{18}O$ and $\delta D$ were collected in polyethylene bottles. Both oxygen and hydrogen isotope ratios were determined using a MAT-253 Stable Isotope Ratio spectrometer, with a precision better than 1, for $\delta D$ and 0.2 for $\delta^{18}O$. Strontium isotope analysis was performed by a Phonixthermo-ionization mass spectrometer, and $SiO_2$ was observed by inductively coupled plasma emission spectrometer Optima-5300 DV [33].

This analysis was performed in the laboratory of the Institute of Earthquake Science, China Earthquake Administration. In addition, pH, electrical conductivity (EC), total dissolved solids (TDS), and temperature were measured in the field. All concentrations were expressed in milligrams per liter (mg/L), except pH and EC. EC concentrations were expressed in μS/cm, and TDS concentrations were expressed in ppm. Physicochemical parameters and the analytical data of major elements and trace elements of the spring waters are shown in Supplementary Materials Tables S1 and S2.

### 3.2. Data Collection and Processing

Five sampling campaigns were performed from 2015 to 2019, but the collection frequency of each hot spring point has a very wide difference. Some springs were collected several times, and some springs were collected only once. In this paper, the average value is used for multi-year data. However, it brings a problem that this kind of approach can complicate data interpretation, especially in areas that have marked seasonal trends, because the annual variance of data could blur the signal of the hydrogeochemical anomaly [34]. Thus, the normalization of data through z-scores was applied for every spring analyzed and highlights the anomalous values for each variable, compared with the geochemical background of different springs. To accomplish this, a z score normalization of data was calculated following the Equation (2):

$$z = (x - m)/s \tag{2}$$

where z is the calculated score, x is the observed value, m is the mean of the whole sampling period, and s is the standard deviation. This calculation was performed for every spring selected in the analysis to highlight the anomalies observed in comparison to what were considered background hydrogeochemical values, including main seasonal trends of each analyzed spring (Figure 2). Normalized values for all the analyzed data were reported in Table S3, Supplementary Material. In this case, the effects of z-score transformation are observable, and analyses of the cations ($Na^+$, $K^+$, $Ca^{2+}$, $Mg^{2+}$) and anions ($F^-$, $Cl^-$, $NO_3^-$, $CO_3^{2-}$, $SO_4^{2-}$, and $HCO_3^-$) all showed very good trends of change, which were not very evident when analyzing the absolute values.

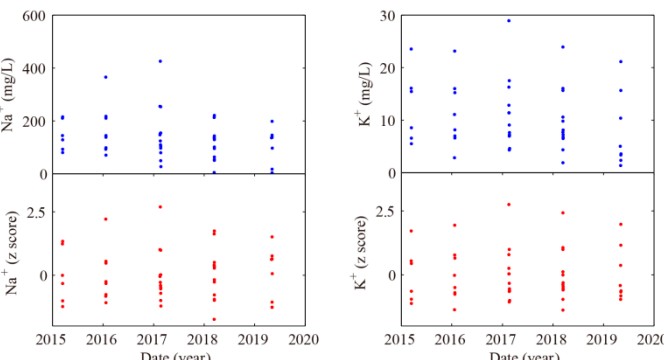

**Figure 2.** *Cont.*

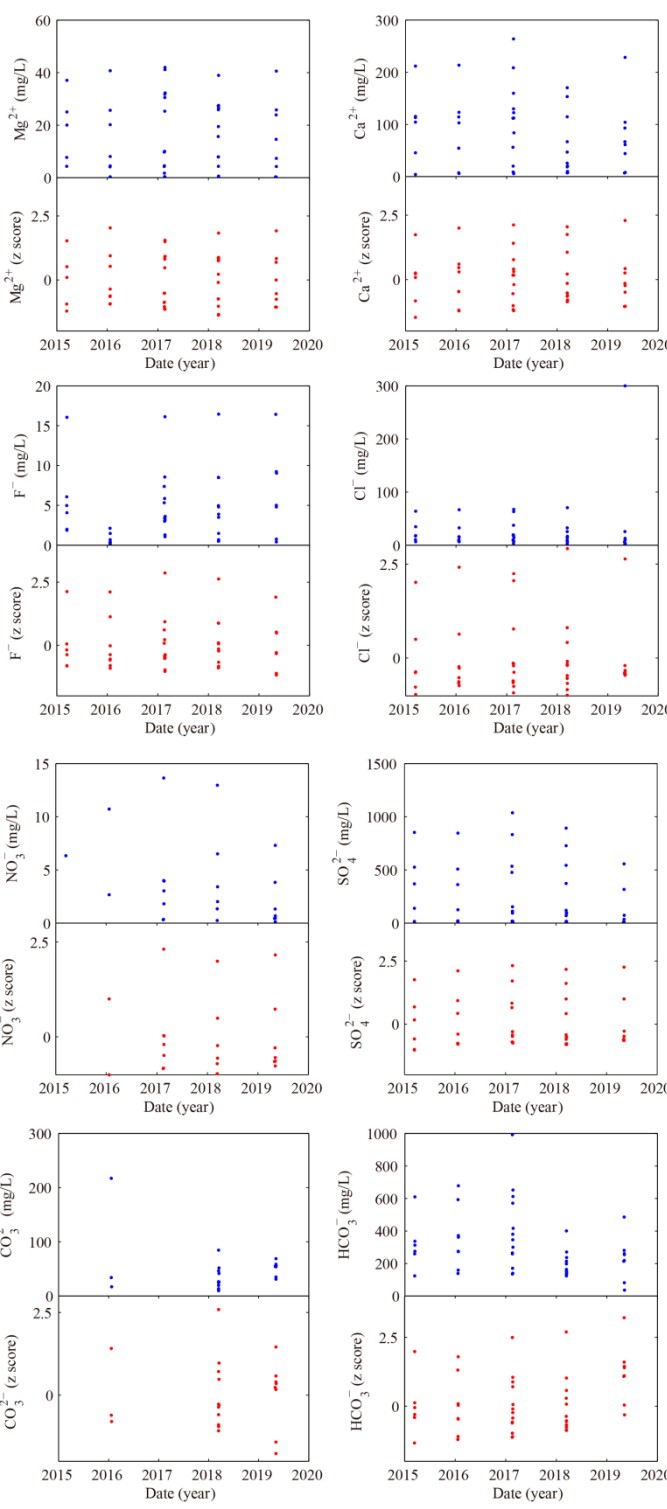

**Figure 2.** Time trend (dates indicated as yyyy) of the absolute values (in mg/L) and of the z scores of the variables $Na^+$, $K^+$, $Ca^{2+}$, $Mg^{2+}$, $F^-$, $Cl^-$, $NO_3^-$, $CO_3^{2-}$, $SO_4^{2-}$, and $HCO_3^-$ from 2015–2019.

## 4. Results and Discussion

### 4.1. Recharge Sources of Hot Spring Water

Hot spring isotopic ratios ranged from $-11.5‰$ to $-8.30‰$ for $\delta^{18}O$ and from $-86.4‰$ to $-59.2‰$ for $\delta D$ (Table 2). In the relational graph (Figure 3) of $\delta D$-$\delta^{18}O$, the full line represents the global meteoric water line $\delta D = 8\delta^{18}O +10$ [35], and the dotted line represents the China meteoric water line $\delta D = 7.54\delta^{18}O + 4.84$ [36]. From Figure 3, it can be seen

that the vast majority of the sampling sites were located close to the meteoric water lines or on two sides, showing that the main source of supply of these springs—for the local atmospheric precipitation and surface water, the deep circulating heating to return to the surface formation of hot springs, and the springs $^{18}$O drift—was not obvious, reflecting that the surface water underground cycle time was shorter. No time for $^{18}$O exchanges with surrounding rock or gas was observed.

**Table 2.** The data of Hydrogen and oxygen isotope compositions, silicon dioxide, strontium, and strontium isotopes from hot springs.

| No. | $\delta D$ | $\delta^{18}O$ | $S_iO_2$ | Sr | $Sr^{86}/Sr^{87}$ |
| --- | --- | --- | --- | --- | --- |
| | (‰) | | (mg/L) | (µg/L) | (‰) |
| S1 | −63.7 | −8.9 | 20.46 | 0.323 | 0.7263 |
| S2 | −73.2 | −10.4 | 69.55 | 0.13484 | 0.727911 |
| S3 | −62.6 | −8.5 | 24.40 | 1.01 | 0.72548 |
| S4 | −59.2 | −8.3 | 24.82 | 1.09 | 0.724765 |
| S5 | −83.4 | −10.9 | 51.79 | - | - |
| S6 | −65.4 | −9.8 | 44.94 | 0.8485 | 0.710358 |
| S7 | −64.2 | −8.9 | 53.71 | - | - |
| S8 | −81 | −11.2 | 103.36 | 0.11033 | 0.710651 |
| S9 | −71 | −10.5 | 85.60 | 2.073 | 0.711759 |
| S10 | −67.3 | −9.2 | 59.92 | 1.05175 | 0.709807 |
| S11 | - | - | - | - | - |
| S12 | −72.8 | −10.6 | 89.24 | 0.233 | 0.720928 |
| S13 | −73.8 | −9.9 | 107.43 | 3.5335 | 0.707867 |
| S14 | −75.4 | −10.9 | 146.16 | 2.85975 | 0.707932 |
| S15 | −77.2 | −10.3 | 153.65 | 0.78525 | 0.71177 |
| S16 | −76.8 | −10.8 | 84.96 | 0.611 | 0.730016 |
| S17 | −77.5 | −10.3 | 21.40 | - | - |
| S18 | −84.2 | −11.3 | 23.11 | - | - |
| S19 | −86.4 | −11.5 | 25.68 | - | - |
| S20 | −75.8 | −9.6 | - | - | - |

"-" represent no data.

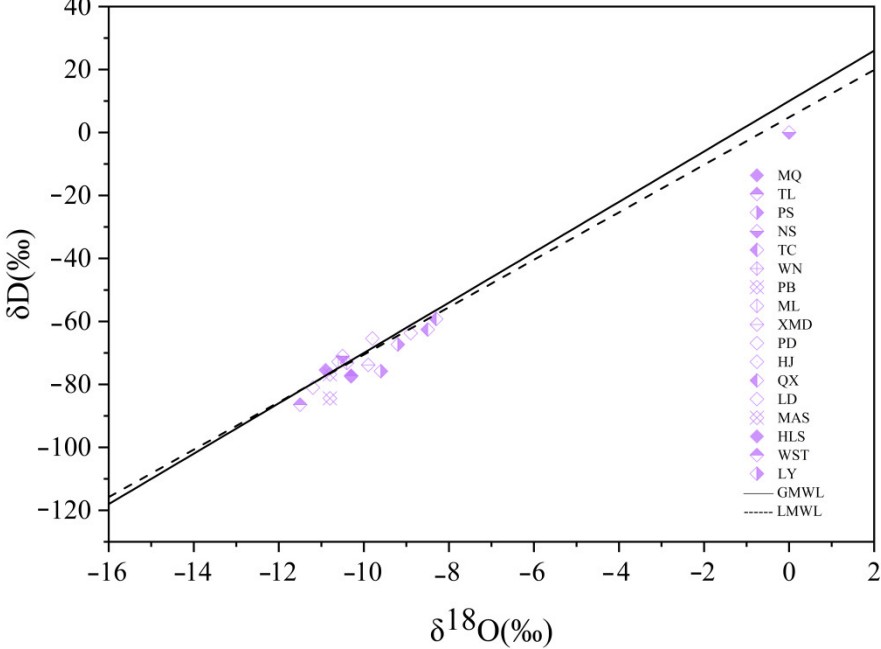

**Figure 3.** Stable oxygen and hydrogen isotope of the 20 hot springs and their correlations with GMWL and LMWL. LMWL: $\delta D = 8.41\delta^{18}O + 16.72$ [5].

The $\delta$D and $\delta^{18}$O of meteoric precipitation decreased with the increase in topographic elevation, according to the relation between the oxygen isotope and the recharging elevation $\delta$D = $-0.03$H—27 [37], $\delta^{18}$O = $-0.0031$H—6.19, $\delta$D = $-0.026$H—30.2 [38]. The recharge elevation was calculated to be 1–2 km. Since $\delta^{18}$O in geothermal water could easily exchange with the surrounding rock, the calculated results of $\delta^{18}$O values were considered biased. In order to reduce the error caused by sampling and testing, the average values of the three methods were used as the recharge area elevation of the hot spring, and we calculated the recharge elevation of the study area to be 1–2 km.

The $\delta$D and $\delta^{18}$O values of precipitation became smaller when the air temperature gradually decreased, and there was a positive correlation with the temperature. The temperature of the recharge area was then estimated from the temperature effect of $\delta$D and $\delta^{18}$O values of precipitation. The temperature of the recharge was estimated in the range of 0.73 °C to 11.86 °C, according to the formula summarized by predecessors: $\delta^{18}$O = 0.521T—14.96, $\delta^{18}$O = 0.176T—10.39, and $\delta$D = 3T—92 [39,40]. The $\delta^{18}$O in the geothermal water was more likely to exchange isotopes with surrounding rocks, so the isotope exchange reaction had little effect on the value of $\delta$D in geothermal water. Therefore, the value of $\delta$D in geothermal water could better reflect the source of geothermal water than the value of $\delta^{18}$O [41]. The average values of the three calculated results were selected in the study, and the temperature of the recharge area ranged from 0.73 °C to 11.86 °C, which was lower than the annual average temperature of Yunnan. It is speculated that this was caused by the high altitude of the recharge area. The result showed that the recharge temperature was approximately proportional to the altitude, which means the recharge temperature corresponding to high altitude was lower than that corresponding to low altitude. For example, the highest recharge elevation at sampling point S19 was 2 km, and the lowest recharge temperature was 0.73 °C, and these findings were consistent with the predicted results.

### 4.2. Origin of Water-Soluble Ions in Hot Spring
### 4.2.1. Origin of Major Elements

The statistical data for the analyzed groundwater samples were presented in Supplementary Table S1. Results of the study revealed that the water temperature values ranged between 22.3 °C and 93.0 °C, with a mean value of 56.1 °C. The value of pH ranged between 6.74 mg/L to 9.02 mg/L, with a mean value 7.51 mg/L, indicating slightly alkaline groundwater in the area. The EC of spring waters of the study areas ranged from 387.0 µS/cm to 1953.0 µS/cm, with an average of 1047.4 µS/cm, showing the presence of dissolved salts in the water that are generally incorporated into the water from geochemical processes such as ion exchange, evaporation, and silicate weathering. The TDS value ranged from 170–4300 mg/L, with an average of 403 mg/L.

We use box plots to describe the concentration of cations and anions (Figure 4). It is observed that the concentrations of cations and anions were ranked in the order $Na^+ > Ca^{2+} > Mg^{2+} > K^+$ and $HCO_3^- > SO_4^{2-} > CO_3^{2-} > Cl^- > F^- > NO_3^-$, respectively. Meanwhile, the data of five ions ($Na^+$, $Ca^{2+}$, $SO_4^{2-}$, $HCO_3^-$, and $CO_3^{2-}$), fluctuated greatly, especially $SO_4^{2-}$ and $HCO_3^-$. The distribution of $Na^+$ was right-skewed, while that of $SO_4^{2-}$ and $CO_3^{2-}$ was left-skewed.

Trilinear plotting systems are utilized to study water chemistry and quality [42]. On conventional trilinear diagrams, sample values for three cations (calcium, magnesium, and alkali metals—sodium and potassium) and three anions (bicarbonate, bhloride, and bulfate) were plotted relative to one another. These ions are usually considered the most common constitutions in unpolluted groundwater. Fundamental interpretations of the chemical nature of water samples were based on the location of the sample ion values. In Figure 5, the water samples were plotted in 1, 2, 3, 4, 5, and 6 blocks. The hot spring samples have different total salinity (TIS), ranging from 2.46 to 22.99 meq/kg. The correlation graph is of $Na^+ + K^+$ vs. $Ca^{2+} + Mg^{2+}$, in which iso-ionic-salinity lines are drawn for reference (Figure 6) [43].

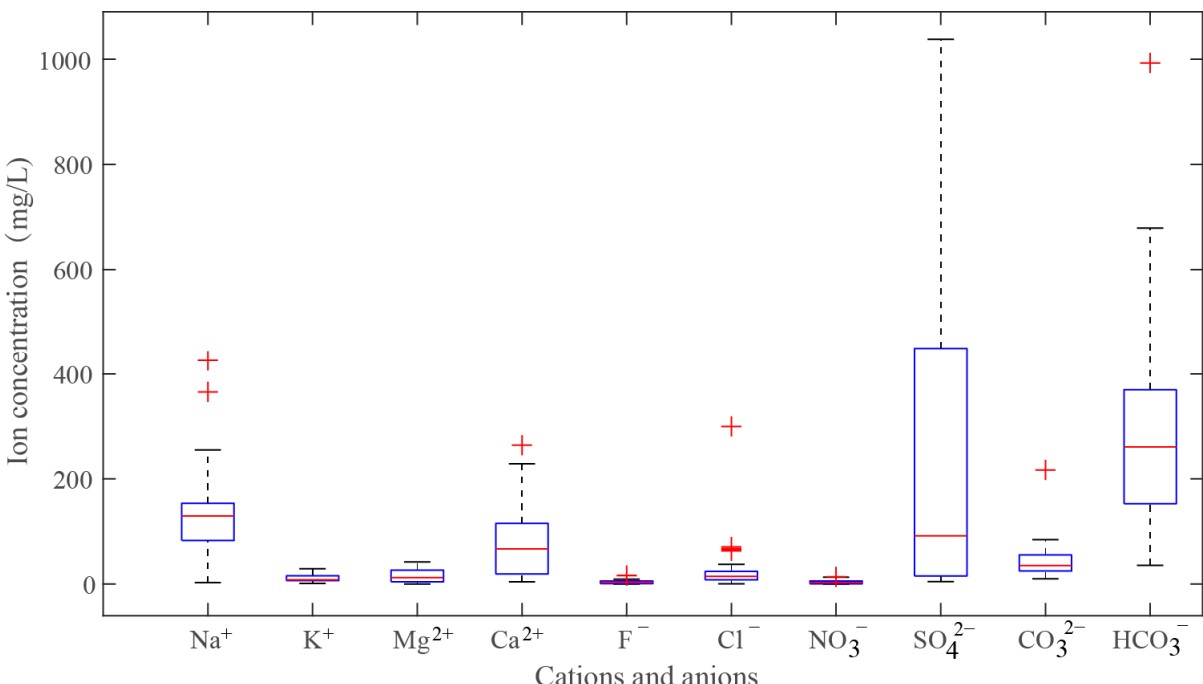

**Figure 4.** Box plots are used to describe the concentration of cations and anions. The width of the box reflects how volatile the data are. The flatter the box, the more concentrated the data. The symbol "+" represents the data outliers, which represents the ion enrichment in this paper.

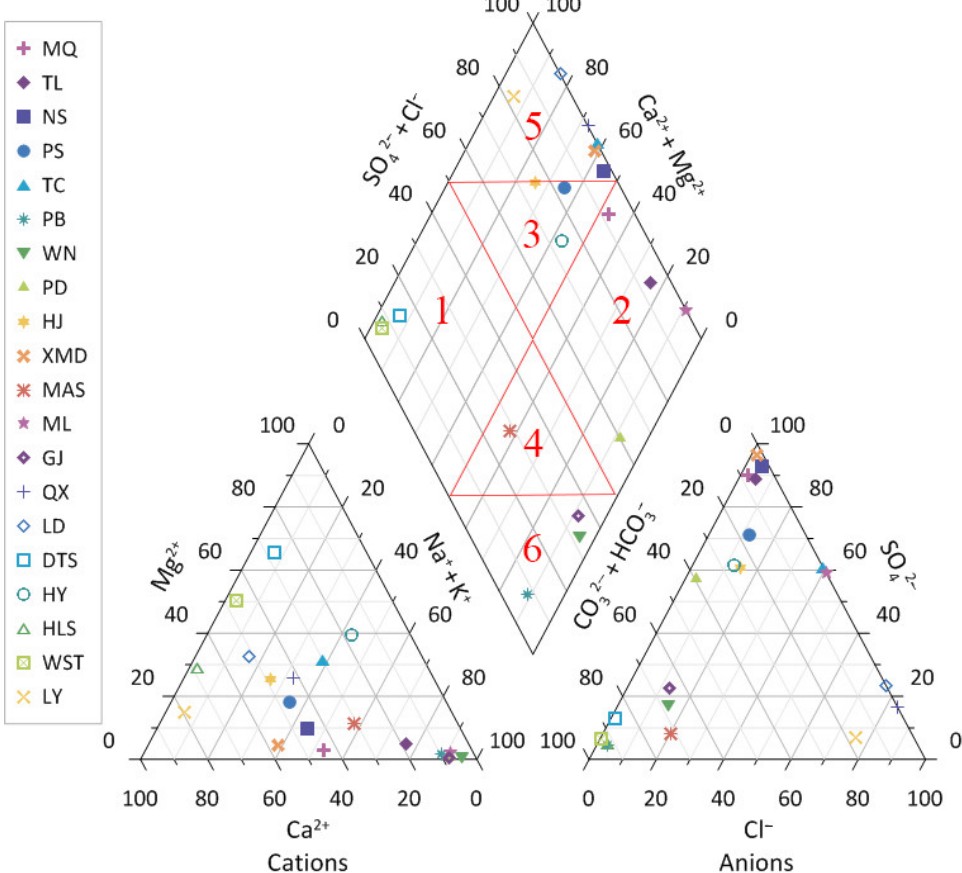

**Figure 5.** Piper diagram showing major ion chemistry of the sampled points.

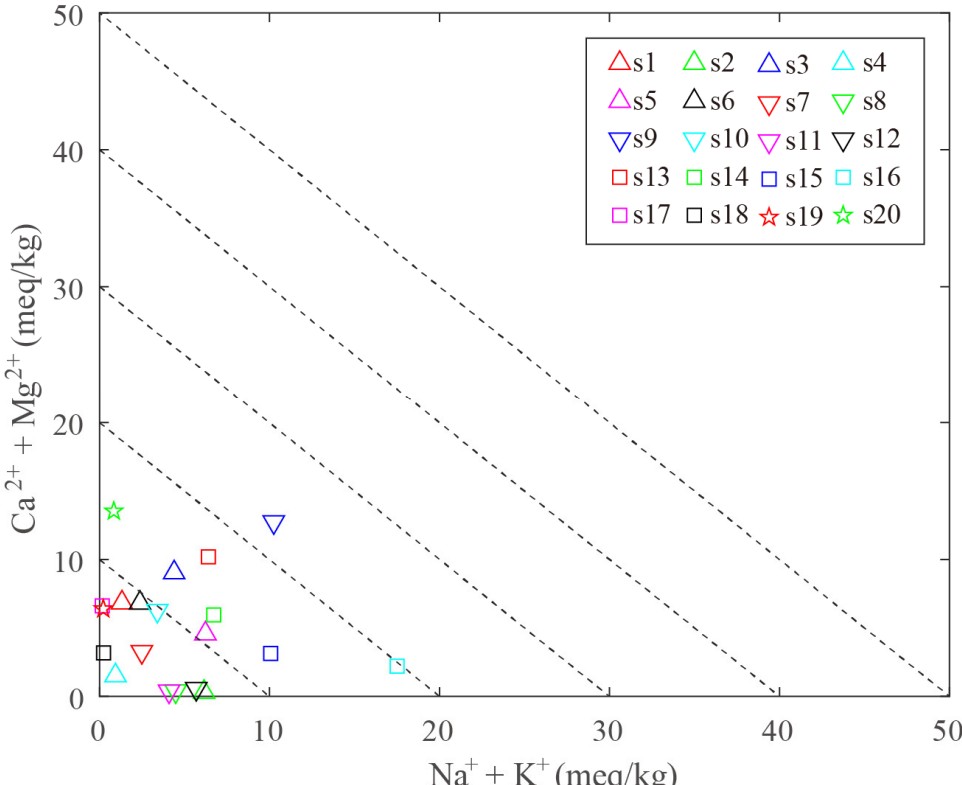

**Figure 6.** Correlation plot of Na$^+$ + K$^+$ vs. Ca$^{2+}$ + Mg$^{2+}$ for the spring water samples in the intersection area of the RRF and the XJF, showing iso-ionic-salinity (TIS) lines for reference.

The type of hydrochemistry of most groundwater from the XJF was Ca-HCO$_3$, accompanied by Ca(Mg, Na)-HCO$_3$ type water. The total salinity (TIS) ranged from 3.42 to 6.77 meq/kg. Unlike the XJF, the main type of hydrochemistry from the RRF was Na-HCO$_3$, accompanied by Na(Ca)-HCO$_3$ type water. The total salinity (TIS) ranged from 2.46 to 13.43 meq/kg. HCO$_3^-$ mainly originates from the dissolution of carbonate in sedimentary rocks and the weathering dissolution of aluminosilicate minerals (albinic, calcium feldspar) in magmatic rocks and metamorphic rocks. Interestingly, the hot springs of S18 and S19 were magniferous. Common sources of magnesium ions in natural water are dolomite, olivine, hornblende, etc. It is also associated with calcium carbonate (CaCO$_3$). Mg(Ca)-HCO$_3$ or Ca(Mg)-HCO$_3$ type water can be frequently seen only in the distribution area of dolomite or dolomitic limestone, which is formed from magnesium-bearing carbonate rocks in the presence of CO$_2$.

In addition, there were sulphuric acid water types. Water samples, such as S9, S13, S14, S15, and S16, with Na(Ca)-SO$_4$, Ca(Na)-SO$_4$, and Na-SO$_4$(HCO$_3$), were mainly distributed in the east of the RRF southern section. The total salinity (TIS) ranged from 12.67 to 22.99 meq/kg, with the higher values. The existence of SO$_4^{2-}$ indicates that the sedimentary rock is interspersed with gypsum (CaSO$_4$·2H$_2$O) or the environment where underground hot water occurs and migrates as an oxidation environment. O$_2$-rich seepage water enters the pyrite-bearing sedimentary layer, and the pyrite dissolves, forming Fe$^{2+}$ and SO$_4^{2-}$. The reaction of sulfuric acid and carbonate generates CO$_2$, which further advances the dissolution of carbonate rock and forms SO$_4$(HCO$_3$) type water. A small amount of pyrite, pyrrhotite, and chalcopyrite is discovered near the eastern section of the RRF [33]. These sulfur-bearing minerals may be oxidized to sulfate by hydrothermal reaction. In addition, they are close to the Pingbian volcano. It is speculated that these hot springs may contain a mixture of acidic high-temperature gases, such as SO$_2$ and H$_2$S, in the deep, in addition to the sulfur-bearing minerals in the dissolved thermal reservoirs, which causes the rise of SO$_4^{2-}$ after dissolution, indicating that the RRF has a deep cutting depth.

Furthermore, the hydrochemical type was Ca-HCO$_3$(Cl) in the site of S20, the high concentration of Cl$^-$ indicated that the water discharged directly from the deep heat reservoir and the degree of mixing or conduction cooling was low, and the water was supplied by the deep water reservoir. The value of the total salinity (TIS) was 14.37 meq/kg. The process could be illustrated in the Equation of (3)–(8).

$$CaCO_3 + H_2O + CO_2 \rightarrow Ca^{2+} + 2HCO_3^- \tag{3}$$

$$2FeS_2 + 7O_2 + 2H_2O \rightarrow 2FeSO_4 + 4H^+ + 2SO_4^{2-} \tag{4}$$

$$2NaAlSi_3O_8 + 3H_2O + 2CO_2 \rightarrow Al_2(Si_2O_5)(OH)_4 + 4SiO_2 + 2Na^+ + 2HCO_3^- \tag{5}$$

$$Na_2CO_3 + H_2O + CO_2 \rightarrow 2NaHCO_3 \tag{6}$$

$$MgCO_3 + H_2O + CO_2 \rightarrow Mg^{2+} + 2HCO_3^- \tag{7}$$

$$Ca\,O_4 \cdot H_2O \rightarrow Ca^{2+} + SO_4^{2-} + H_2O \tag{8}$$

The main hydrochemical types indicated that the major ions dissolved in these spring points were related to the properties of the exposed surrounding rock. The leaching of calcite, dolomite, and other minerals in surrounding rock was the main source of ions.

### 4.2.2. Origin of Trace Elements

Trace elements in hot springs can reflect the water–rock reaction degree of groundwater to a certain extent. In this paper, the presence of 27 trace elements was observed in 19 sampling sites, except for S11, which included Hg, As, Ag, B, Li, Be, Ti, V, Mn, Cr, Co, Ni, Cu, Zn, Mo, Cd, Sb, Ba, T1, Pb, Th, U, Sr, Sn, Fe, Al, and Se. Table S2 indicates the concentration of the trace elements analyzed in this study.

The enrichment factor [44,45] was a double normalization calculation method, as used for comparing trace element patterns among different types of waters. It selects elements that meet certain conditions as reference elements (or standardized elements). In this paper, the enrichment factor is chosen to describe the enrichment of the trace elements, and the elements that are ubiquitous in the Earth's crust and have less anthropogenic pollution sources and good chemical stability are selected as the reference elements of the calculation formula, according to Equation (9),

$$EF_i = (C_i/C_R)_w / (C_i/C_R)_r \tag{9}$$

where subscripts w and r relate to water and rock, respectively. Alkali-rich porphyry in the southern section of Ailaoshan-RRF belt, Yunnan Province, and crustal element Ti are selected as reference elements in water and rocks, respectively. The results are shown in Figure 7.

The $EF_i$ of Be, Cr, Co, Ni, and Cu were relatively higher than others, and Ba was relatively lower than others. The $EF_i$ of Cr, Co, Cu, and Ni, in S17, S19, and S20 hot springs, mainly attributed to the XJF, which was relatively higher than others. It was indicated that there were mainly sulphide salts or carbonate in the XJF, which had high solubility and strong migration ability. Ions were enriched under the condition of strong oxidation. Furthermore, it was compatible with the local high altitude and underground runoff conditions. High concentration areas of Fe$^{2+}$ were mainly distributed in S1 and S20 hot springs, which are mainly attributed to the pyrite dissolution of the lignite of surrounding rocks in the hot spring.

### 4.2.3. Strontium Isotope Analysis

Strontium isotope composition in hot spring water can reflect the lithologic characteristics of the strata through which it flows [46]. Therefore, $^{87}$Sr/$^{86}$Sr values from different lithologic types were used in this paper (Figure 8 and Table 2). The $^{87}$Sr/$^{86}$Sr values from carbonate and sulfate weathering sources were about 0.708000. The $^{87}$Sr/$^{86}$Sr values of

aluminum silicate weathering sources generally ranged from 0.716000 to 0.720000. As can be seen from Figure 5, the S1, S2, S4, S3, and S12 hot springs belonged to the silicate mineral weathering, the S13, S14 hot springs belonged to the carbonate mineral weathering, and the remnant springs were found between carbonate and silicate mineral weathering. They were formed by Sr-bearing rocks in the crust, consistent with the water interaction with surrounding rocks, during the deep circulation of atmospheric precipitation.

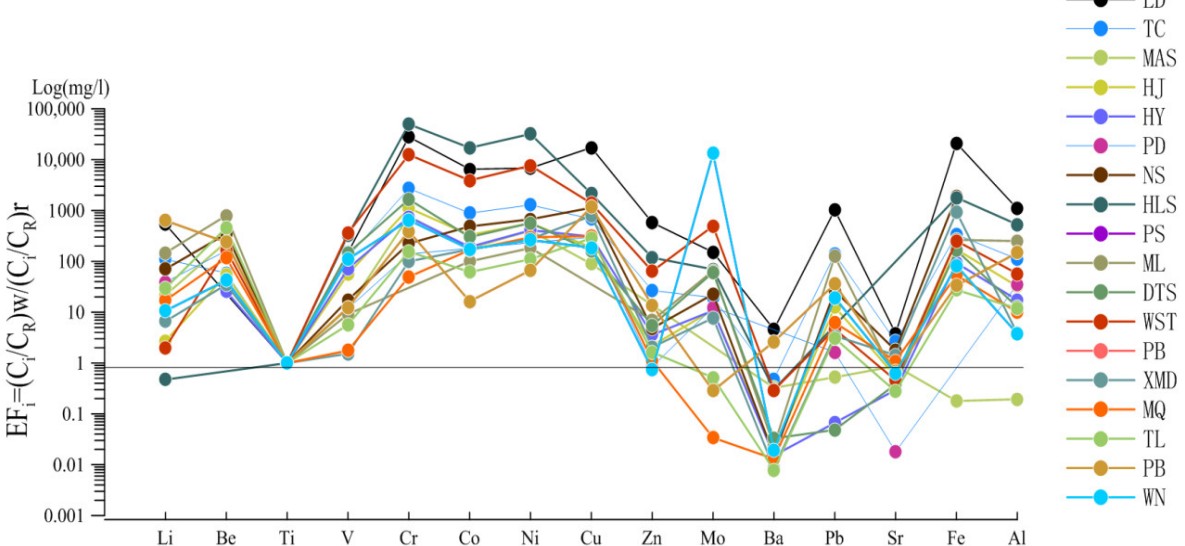

**Figure 7.** Trace element distribution, in terms of enrichment coefficient normalized to Ti, for the waters of the intersection of the RRF and the XJF.

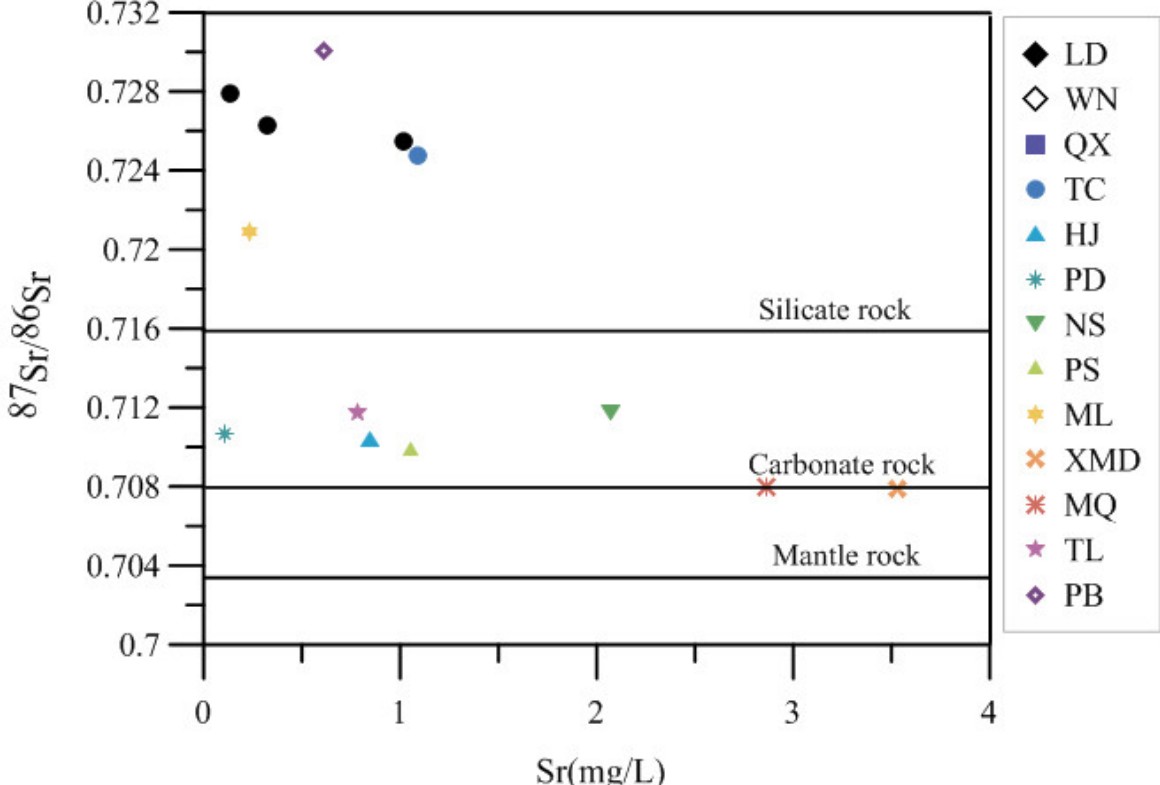

**Figure 8.** Strontium isotopic composition of the 20 hot springs in the study area.

### 4.3. Water Rock Interaction of Hot Springs

#### 4.3.1. The Water–Rock Reaction Equilibrium

The Na-K-Mg triangle diagram [47] was used extensively to evaluate the water–rock balance state, and it distinguishes different types of water samples. Figure 9 showed that the S2, S8, S11, and S16 water samples were felled below the part of the equilibrium water area, which was rich in $Na^+$ and lacking of $Ca^{2+}$ and $Mg^{2+}$, and the main hydrochemical type was $NaHCO_3$. The water temperature of the four hot springs ranged from 66 °C to 80.95 °C, and the equilibrium temperature was between 140 °C and 180 °C, indicating that the water samples came from a relatively hot environment, and sufficient water–rock reaction was carried out with the surrounding rock during the deep circulation of hot water. However, it may have been combined with different degrees of shallow cold water during the ascent. Other hot spring samples were felled in the immature water area, indicating that the equilibrium temperature of the water–rock reaction was not high, the reaction between groundwater and the surrounding rock was insufficient, or that it could also be due to cold water dilution. This indicated that most of the hot springs in the study area were supplied by meteoric precipitation with a few using mixed cold water. It also indicated that RRF had a large cycle depth and high fracture opening degree, and the stress was not easy to accumulate.

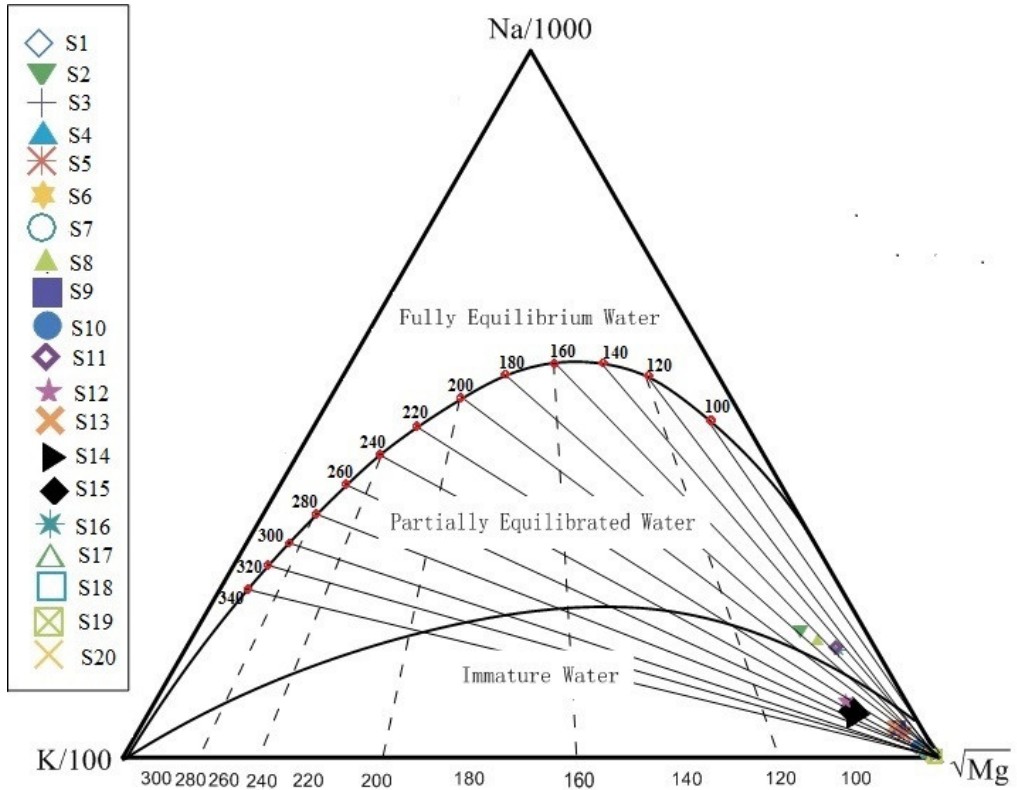

**Figure 9.** Distribution of aqueous samples on the Na/1000-K/100-Mg1/2 ternary diagram.

#### 4.3.2. Mixing with Shallow Groundwater or Surface Water

Isotope δD, with $Cl^-$ and a Na-K-Mg ternary diagram, can be used to analyze whether mixing occurs [48] (Figures 9 and 10). When analyzing the relationship between $Cl^-$ and δD, the study is divided into two groups for discussion, according to the geographical distribution (S1–S7, excluding S5 because S5 is not on the same side of the RRF as other springs): one group is the western segment of the RRF, while the other group is the intersection area of the RRF and the XJF (S8–S19, since the concentration of $Cl^-$ in S20 is much higher than that at other points, which are not to be discussed together here).

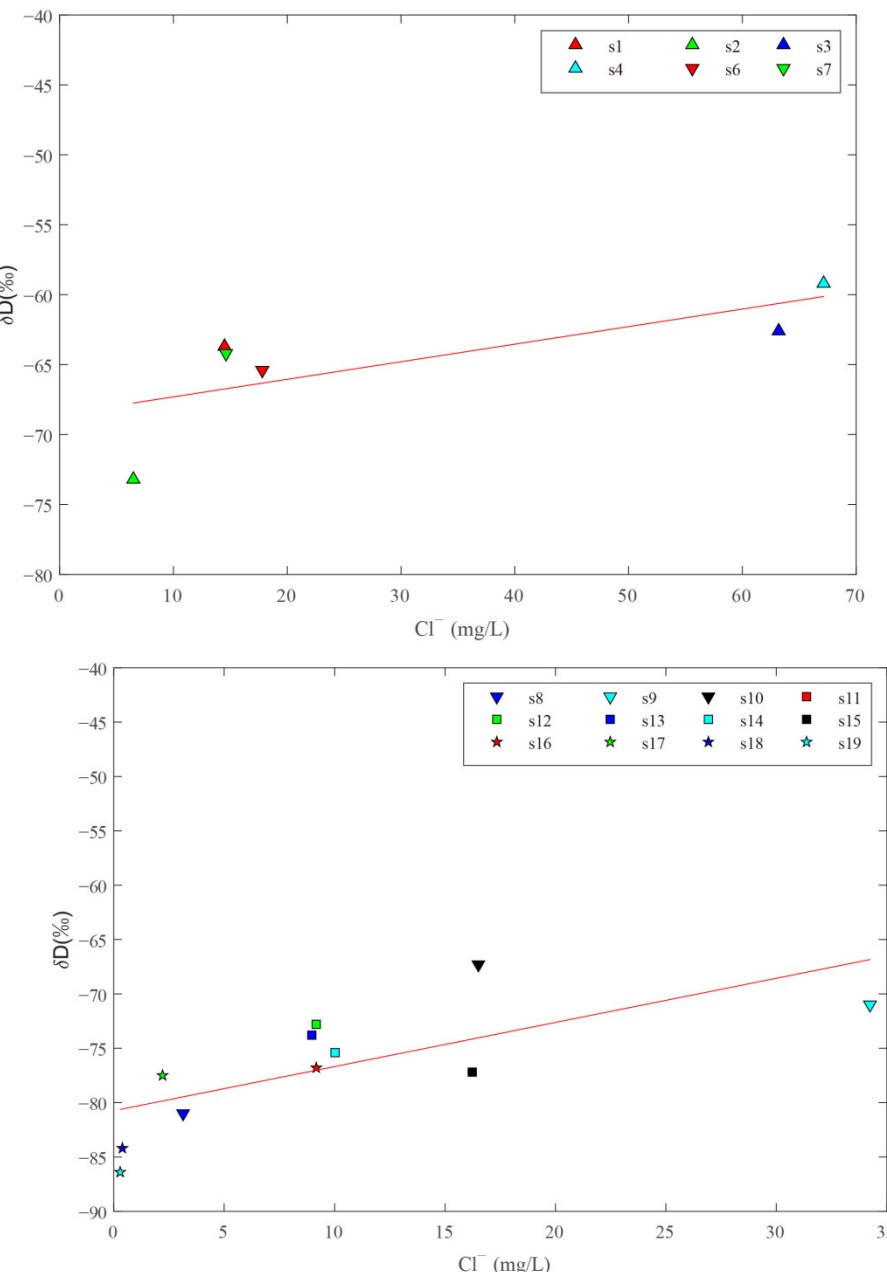

**Figure 10.** Relations of δD with Cl-, for various waters, at the intersection of the RRF and the XJF.

As shown in Figure 10, correlation between δD and $Cl^-$ are so good because they have the same local precipitation origin. S1, S2, S4, S6, S10, S12, S13, S14, S16, and S17 springs are on, or above, the mixing line but have higher ion concentrations than shallow groundwater due to strong evaporation. The mixing trend can also be verified from the Na-K-Mg ternary diagram, which is proposed by Giggenbach to estimate the deep reservoir temperature and equilibrium state of water samples. In Figure 9, an obvious linear relationship can be observed among S1, S2, S4, S6, S10, S13, S16, and S17 springs. Therefore, it is concluded that mixing with shallow groundwater is definitely occurring for geothermal fluid when upwelling to the surface. This result is also consistent with the results discussed in Section 4.3.2.

### 4.3.3. The Ratio of Cold and Hot Water

According to the Na-K-Mg triangle diagram, the hot spring water in the study area was mixed with shallow cold water in the upwelling process. There were 18 hot spring

sites that were involved in the calculation (except S11 and S20) for lacking silicon values. The silicon–enthalpy equation method was used to calculate the mixing ratio of hot and cold water. Assuming that the ratio of mixing cold water was denoted as X, the relationship between initial enthalpy and initial content of silica of underground hot water, as well as final enthalpy and concentration of silica of spring water, is [49], $ScX_1 + S_h(1-X_1) = Ss$, $SiO_{2c}X_2 + SiO_{2h}(1-X_2) = SiO_{2s}$, where $S_h$ is the initial enthalpy of hot water, Sc is the enthalpy of near-surface cold water, and the average temperature of 21 °C in Southeast Yunnan is adopted in this paper [50]. $SiO_{2h}$ is considered the initial concentration of $SiO_2$ in deep hot water and is the concentration of $SiO_2$ in near-surface cold water, and 11 mg/L concentration was used in this paper [50]. $SiO_{2s}$ is the concentration of $SiO_2$ in hot spring water. The relationship between temperature, enthalpy of hot water, and the concentration of $SiO_2$ [49] is shown in Table 3. The origin software was used to draw the curves of $X_1$ and $X_2$, respectively, to obtain the relationship diagram of the mixing ratio of hot and cold water, in hot springs, in the study area (Figure 11). The horizontal coordinate, corresponding to the intersection point of the curves, was considered as the proportion of mixing cold water X. The coordinate corresponding to the intersection point was the initial enthalpy of deep hot water: namely, the heat storage temperature. The calculation results showed that S7, S9, S10, S12, S13, and S14 had no intersection point, which was probably because the deep hot water needed to rise to mix. There could be a loss of steam, or non-adiabatic conduction heat loss, including the mixing of hot and cold water, thereby losing heat while implementing the effects of $SiO_2$ concentration.

**Table 3.** The relationship between temperature, enthalpy, and $SiO_2$ content of hot water.

| Temperature/°C | 50 | 75 | 100 | 125 | 150 | 175 | 200 | 225 | 250 | 275 | 300 |
|---|---|---|---|---|---|---|---|---|---|---|---|
| Enthalpy/(4.1868J/g) | 50 | 75 | 100.1 | 125.4 | 151 | 177 | 203.6 | 230.9 | 259.2 | 289 | 321 |
| $S_iO_2$ Content/(mg/L) | 13.5 | 26.6 | 48 | 80 | 125 | 185 | 265 | 365 | 486 | 614 | 692 |

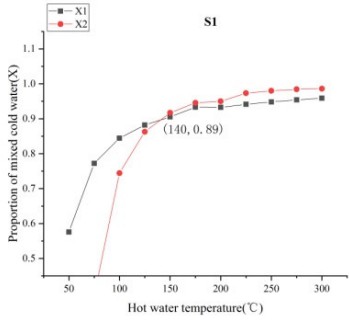
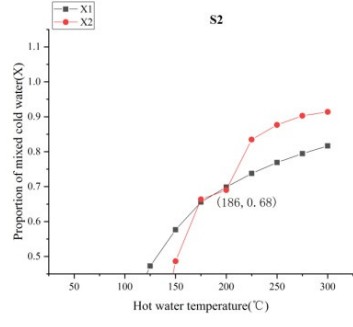
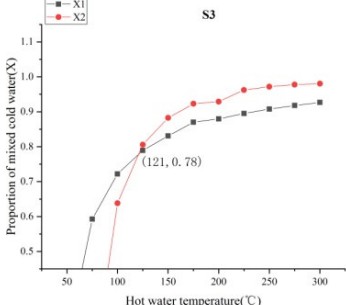
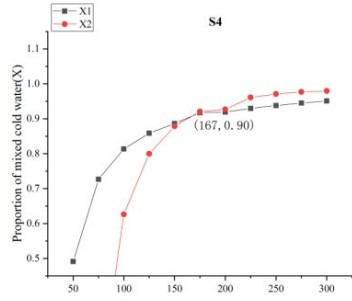

**Figure 11.** *Cont.*

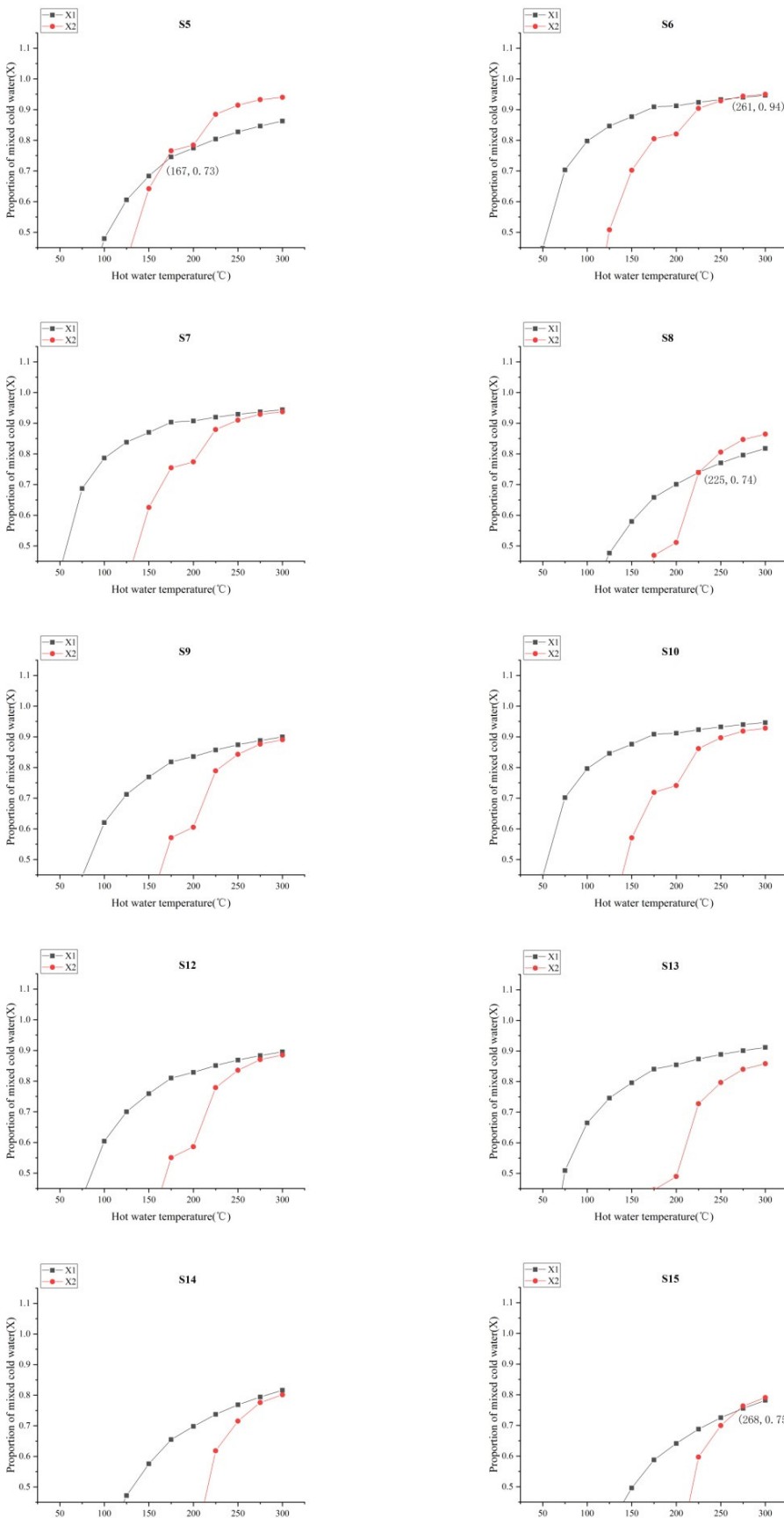

**Figure 11.** *Cont.*

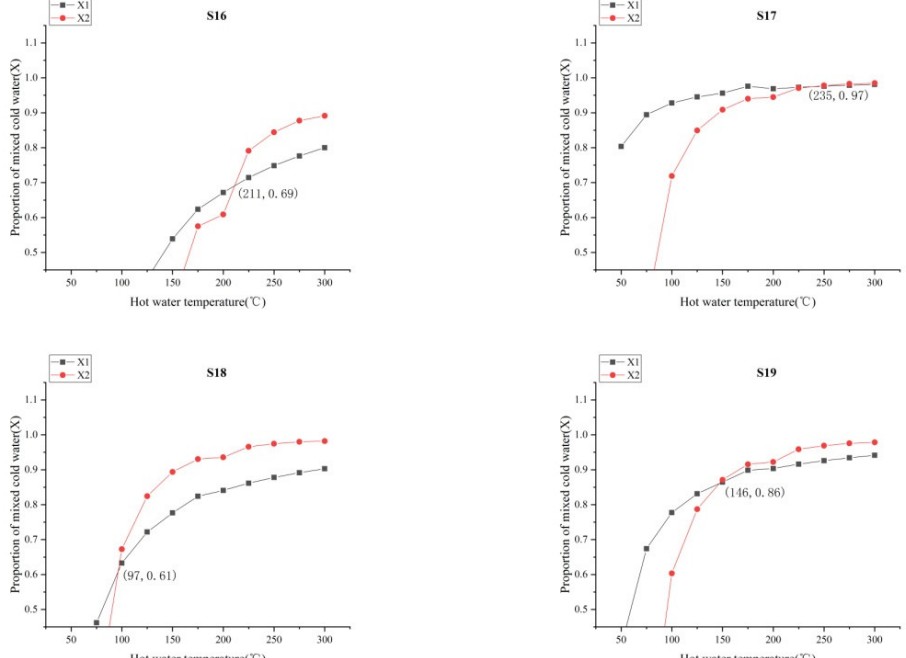

**Figure 11.** Diagram of hot spring hot and cool water mixing ratio in the study area, based on the silicon enthalpy equation method. $X_1$ is the mixing ratio of each enthalpy value calculated at various assumed initial temperatures of hot water; $X_2$ is the mixing ratio of each $SiO_2$ concentration calculated at various assumed initial temperatures of hot water; S1−S19 represents the number of hot spring samplings.

The heat storage temperature range of 12 sampling points (S1, S2, S3, S4, S5, S6, S8, S15, S16, S17, S18, and S19), calculated by the silicon–enthalpy equation, was 97–268 °C, the average temperature was 185.3 °C, the cold water mixing ratio was 61–97%, and the average mixing ratio was 80%. The result is consistent with the finding derived in Section 4.3.1.

4.3.4. Geothermal Reservoir Temperature and Circulation Depth

Geothermal reservoir temperature [51,52] is a key parameter to study the genetic types of heat storage, which evaluate the geothermal energy potential. In the study, we used the geothermal temperature scale to calculate. The silica geothermal temperature scale, based on the theory that the silica content in geothermal fluids depends mainly on the solubility of quartz in water at different temperatures and pressures, is the earliest and most commonly used geochemical temperature scale. Data of the reservoir temperature were evaluated, according to Equation (10):

$$T = 1309/(5.91 - \log (C_{SiO2})) - 273.15 \tag{10}$$

$C_{SiO2}$ indicated the concentration of $SiO_2$ in the water, and the results are shown in Table 2. The reservoir temperature of spring samples, at the intersection of the RRF and the XJF, is mainly 64.3–162.7 °C, and the average temperature is 107.8 °C.

Compared with the heat storage temperature obtained by the silicon–enthalpy model, the calculation result of $S_iO_2$ geothermal temperature scale was smaller because the method doesn't take into account the fact that the hot water will be mixed with cold water as it rises, so the reducing concentration of $S_iO_2$ was leading up to the low calculation results. The silicon–enthalpy model is based on the relationship curve between quartz solubility and temperature, which reflects the maximum temperature of the deep hot water unit before mixing. Therefore, the silicon–enthalpy model can better reflect the real deep heat storage temperature of the geothermal system. For sampling points S7, S9, S10, S11, S12, S13, S14, and S20, the heat storage temperature was calculated using the $S_iO_2$ geothermal

temperature scale. For sampling points S1, S2, S3, S4, S5, S6, S8, S15, S16, S17, S18, and S19, the average value of the silicon–enthalpy equation method and the silicon–enthalpy diagram method was the heat storage temperature.

Data of circulation depth were evaluated, according to the geothermal gradient and hot spring geothermal reservoir temperature, and the formula is as follows Equation (11).

$$Z = Z_0 + (T - T_0)/T_{\text{grad}} \tag{11}$$

$Z$ is the circulation depth (km); $Z_0$ is the depth of the constant temperature zone (km); $T$ is the reservoir temperature (°C); $T_0$ is the temperature of the constant temperature zone (°C) and, namely, the local average temperature; $T_{\text{grad}}$ is the geothermal gradient (°C/km) reflecting the geothermal change, per one kilometer, of the place below the constant temperature zone [53]. By taking reference from previous studies on the groundwater in some areas of Yunnan Province, the geothermal gradient $T_{\text{grad}}$ was assumed as 20 °C/km, the annual mean temperature $T_0$ assumed was 15.8 °C, and the depth $Z_0$ of the constant temperature zone assumed was 30 m [53]. The final circulation depth of the intersection of the RRF and the XJF was about 1.1−7.2 km as calculated.

### 4.3.5. Mineral Saturation States

The saturation index (SI) is a part of the parameters used to obtain water–rock interaction information, which determines the reaction state between minerals and aqueous solutions [54], and can be calculated by Phreeqc software. Results are presented in Figure 12. Nearly all spring water samples, with the exception of sample S16, were supersaturated (SI > 0) with respect to goethite and hematite in sampling temperatures, suggesting that the surrounding rock of the hot spring contained a significant amount of iron ions. Most of the groundwater samples were in equilibrium with gibbsite and barite (SI ≈ 0). Other spring water samples were not saturated (SI < 0). However, SI, with respect to hausmannite, melanterite, alunite, jarosite—K, pyrolusite, and manganite, varied greatly in each hot spring water. This phenomenon may reflect the difference in the surrounding rock characteristics, indicating that ionic equilibrium was not established between water and rock, whereas the dissolution was still continuing or hot water was mixed with cold water. This conclusion is consistent with the equilibrium state of the water–rock reaction, as discussed in Section 4.3.1.

### 4.4. Relationship between Spatial Distribution of Hydrogeochemical Characteristics and Seismicity at the Intersection of the RRF and the XJF

According to geographical distribution and fault location, hot spring samples in the study area were divided into four segments, from west to east, according to longitude: first segment (S1–S7), second segment (S8–S11), third segment (S18–S20), and fourth segment (S13–S15). Among them, the third segment was located in the XJF, and the other three segments were distributed on the RRF. S17 was in the intersection of Jianshui fault zone (JSF) and the western branch of XJF, S12 was in the Dien Bianfu fault zone (DBFF), and S16 was in the Mengzi–Nanxihe fault zone (Figure 1). Therefore, those three hot spring samples were not discussed here. We projected the water temperature, earthquake magnitude, focal depth, heat storage temperature, cycle depth, and slip rate of all hot spring points on different fault surfaces onto the same figure (Figure 13). We found that: (1) the water temperature, geothermal reservoir temperature, and circulation depth of hot spring samples (the first, second, and fourth segments) on the RRF showed an increasing trend, from west to east, according to longitude, and their average values were higher than those (the third segment) on the XJF, respectively. There was a significant difference between heat storage temperature and circulation depth. It revealed that the hot water was obviously controlled by the fracture and cutting depth. The cutting depth of the XJF was lower than that of the RRF. Previous studies agreed with the RRF as an active block boundary, which may cut through the entire crust [22], while the XJF was only cut deep within the crust [55], which supported our results. (2) In addition, the water temperature of hot springs on the

XJF showed a decreasing trend from north to south, while the heat storage temperature and circulation depth did not change much, indicating that the hot springs on the XJF were shallow circulation, and the hot water may be mixed with more cold water during the rising process. (3) From the perspective of the number of earthquakes, magnitude, and focal depth, the second section was obviously different from the other three sections in terms of fewer earthquakes, larger earthquake magnitude, lack of small and medium earthquakes, and shallow focal depth. It was speculated that this may be related to the control of granite strata, which was consistent with the distribution of granite (Figure 1c).

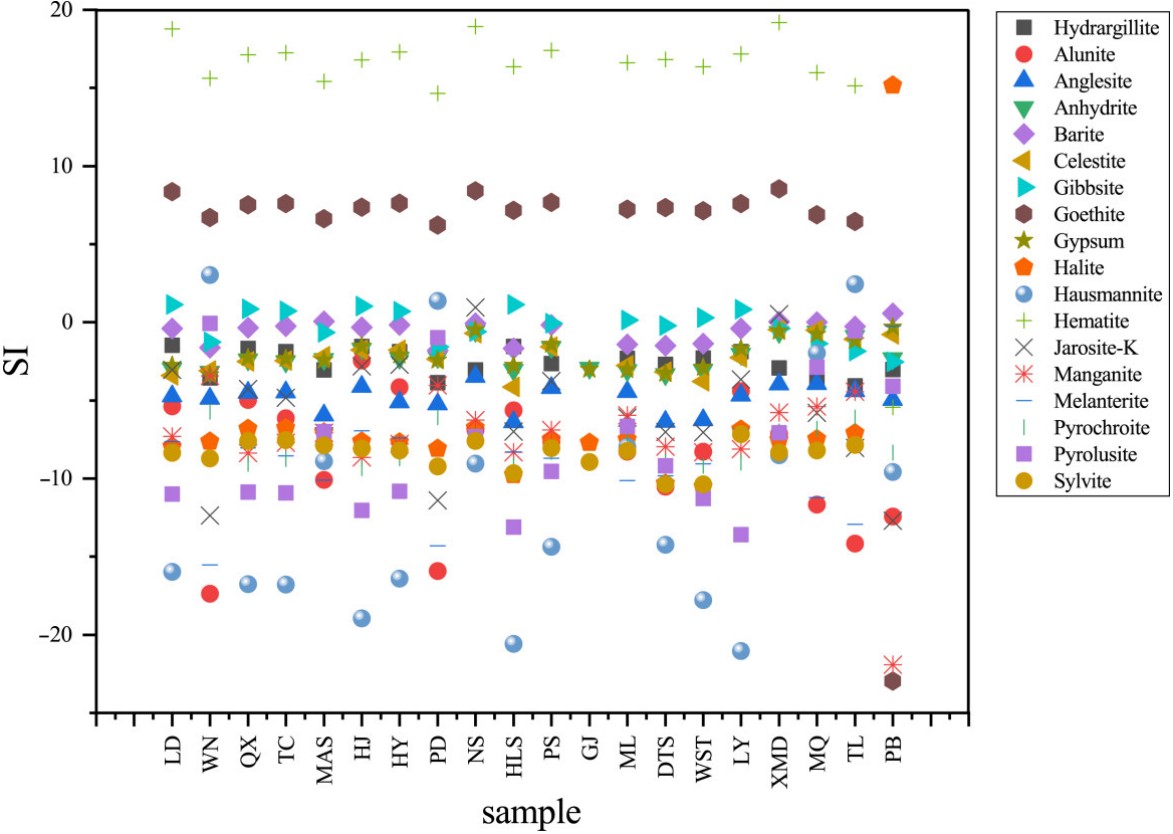

**Figure 12.** Saturation indices values of groundwater samples, with respect to minerals.

According to the circulation depth and heat storage temperature, it can be judged that the second and fourth sections are geothermal high-value areas with strong water–rock reaction of groundwater. However, their seismic activity was quite different, with segment 4 having more frequent small earthquakes than segment 2. It was inferred that the distribution, heat storage temperature, and circulation depth of hot springs were likely associated with seismic activity. The second segment intersected the west branch of the XJF, and the fourth segment intersects the east branch of the XJF. The thermal reservoir temperature and circulation depth of the fourth section are larger than those of the second section. Due to the interaction between underground hot water in high geothermal areas, the reaction between underground hot water, including intensified surrounding rock, increased pore pressure, fluid expansion under heat, and increasing weakening degree of fault [56], i.e., the effective stress on the fault, was not easy to accumulate, and the energy was released in the form of fault slip or a small earthquake, but the possibility of the strong earthquake was less. Instead, in the area with low water temperature, circulation depth, and heat storage temperature, the reaction degrees of underground hot water and surrounding rock were weak; hence, the strength of the weakening effect of groundwater on faults was slow. If the fault lock-in degree in this area was strong, the accumulated tectonic stress would not be so easy to release. When the accumulation of energy reaches the limit, the rock fault will become unstable, and a big earthquake may occur. It was

indicated that the eastern branch of the XJF was more prone to accumulation of stress than the western branch; hence, it was speculated that the western branch had a higher risk of large earthquakes than the eastern branch. This conclusion was supported by the observed resistivity of the eastern and western branches of XJF from magnetotelluric studies [54,57].

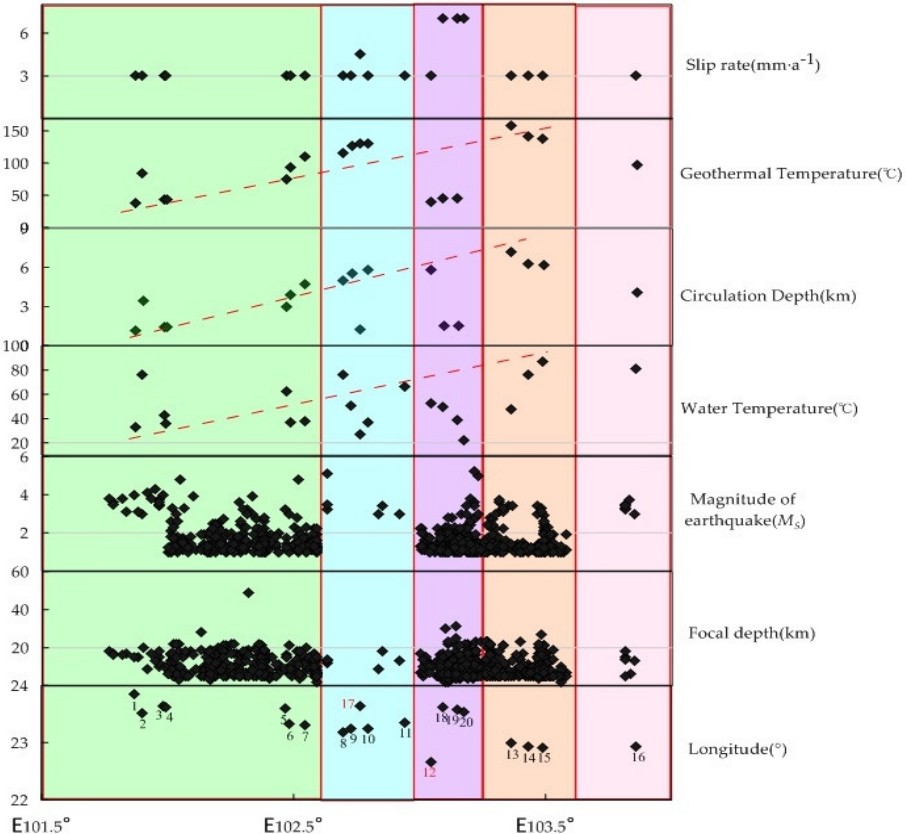

**Figure 13.** The spatial distribution of temperature, circulation depth, slip rate, magnitude, and focal depth at the intersection of the RRF and the XJF were studied. In order of longitude, from west to east, green is S1–S7 (RRF), blue is S8–S11 (RRF), rose is S18–S20 (XJF), orange is S13–S15 (RRF), and pink is S16 (RRF). S17 is the intersection of the JSF and the XJF, S12 is in the DDFF, and S16 is in the MZ-NXHF, so this figure is not discussed. Earthquake time ranged from 1 January 2014 to 31 December 2020, Magnitude ≥1.

In particular, although the third segment belonged to the geothermal low-value areas, it also showed the characteristics of strong seismicity. On the one hand, this was because the XJF activity rate was large, increasing the fault slip, circulation, shallow depth, and fault stress release in the form of small earthquakes. Another reason to speculate was that the faults were developed and provided a good channel for the upward migration of deep fluids in the intersection area, which played an important role in promoting the gestation and occurrence of earthquakes. More and more geophysical studies supported the conclusion that there were high conductivity and low velocity layers in the middle and lower crust in the intersection area of the RRF and the XJF [58,59]. In addition, studies on gas geochemical characteristics have also shown that the geothermal high-value area almost coincides with the mantle-derived helium release high (high 3He/4He) area [60]. Therefore we favor that the intensity of hydrothermal activity on the fault zone controls the seismic activity to a certain extent.

### 4.5. Hydrogeochemical Cycle Model

The geothermal hot spring in the intersection area of the RRF and the XJF was a typical high-temperature geothermal system with deep fractures (Figure 14). The high

porosity, caused by the fault structure, allowed water to penetrate and maintain good connectivity [59]. It was inferred that faults are tunnels of these groundwaters in the crust, and during the deep tectonic cycle, the groundwater enters the RRF and the XJF junction zone through the water-conducting fracture zone along the mountain and river terraces and the surrounding fractures. Due to of the crust at the intersection, the RRF and the XJF have high conductivity and low-velocity layers, and they are rich in deep-source fluids. These confined deep-source fluids are discharged on the surface through fractures or cracks (pores) in the lithosphere or the upper crust, thus promoting the upwelling of the asthenosphere. Zhou [60] found that the $^3$He/$^4$He ratio of hot spring gas geochemistry, in the Jinshajiang-RRF (0.04–0.62 Ra), indicated that the helium release from the mantle source gradually increased along the fault zone, from north to south, and the maximum value remained at only 7.5% in the southern segment of the RRF in southeast Yunnan. The 3-D Magnetotelluric model [58] suggested a significant conductor in the upper mantle northeast of the RRF, which extended upward into the crust, and required a melt fraction of up to 3%, which may be due to melt/fluids derived from the mantle. In addition, Li [57] proved that there may be an interaction between mantle and crustal material in the XJF south of 26° N, and the interaction between mantle and crustal material may elevate regional heat flow, so fluid may be uplifted along deep faults. Previous studies supported that the heat source of the high-temperature hydrothermal system was mainly caused by a deep tectonic cycle and crustal radioactivity, and a small amount of a mantle magmatic heat source was involved. When the groundwater circulates downward to about 7.2 km, is heated by deep magmatic heat, granitoid radiative heat, or a sliding friction heat source of the RRF and the XJF, and the water temperature is up to 138 °C, then the water–rock reaction would occur with surrounding rocks at different depths (i.e., granite, igneous rock, etc.) under certain temperature–pressure conditions. Due to different extents of reactions, partially equilibrated water and immature water were produced, which could then be mixed with the cold surface water or shallow groundwater when ascending to the surface ground, and finally, they were exposed on the Earth's surface as a hot spring. We noted that, if the circulation depth of hot spring water is deeper, the water will weaken the fracture more, and the strength of the fracture will be lower, thus affecting the stress state and seismicity of the fracture. It follows that the research results of this paper provide a new method for seismic zoning and the determination of potential source areas.

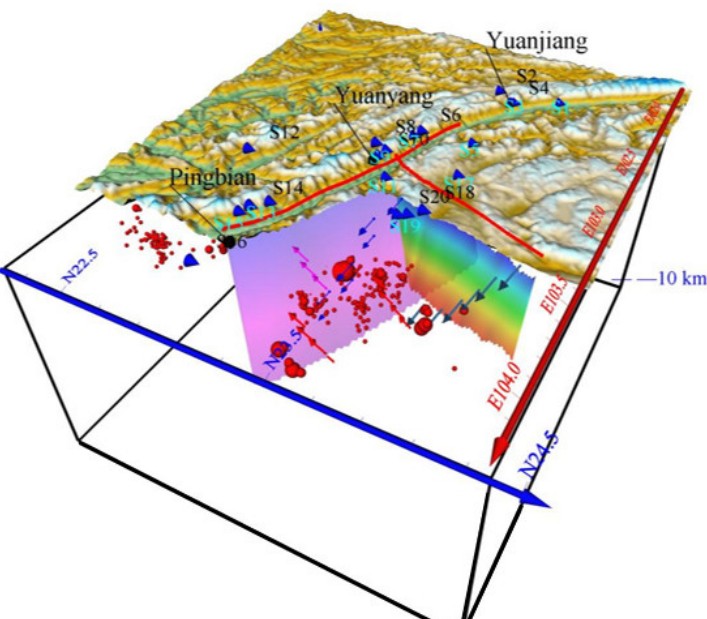

**Figure 14.** Circulation model of geothermal fluid at the intersection of the RRF and the XJF.

## 5. Conclusions

Using a large amount of hydrochemical data from 20 hot spring sites, the detailed mechanisms and processes of geochemical changing intersections of the RRF and the XJF were comprehensively described in terms of regional groundwater flow systems. The results of the study suggested that:

(1)   The temperature of the recharge area ranged from 0.73 °C to 11.86 °C. The hot spring water intersection of the RRF and the XJF was recharged by atmospheric precipitation, recharged elevations ranged from 1.1 to 2 km, and it was positively correlated with the elevation of the hot spring sampling sites.

(2)   Hydrochemical types were mainly controlled by aquifer lithology, in which sodium bicarbonate and sulphuric acid water gathered mainly in the RRF, while calcium bicarbonate water gathered mainly in the XJF. The temperature range was inferred from an equation, based on the $SiO_2$ concentration and chemical geothermal method, as 64.3–162.7 °C. The temperature range was inferred from an equation, based on the silicon–enthalpy equation method, as 97–268 °C, and the cold water mixing ratio was 61–97%. The circulation depths for springs were estimated in the range from 1.1 to 7.2 km. Our data showed that the circulation depth of the RRF was deeper than that of the XJF, and it was mainly concentrated in the second and fourth segments of the RRF.

(3)   It was speculated that the hot water intersection of the RRF and the XJF was obviously controlled by the fault and the cutting depth of granite.

(4)   Relationship discussed between geothermal anomaly and earthquake activity had a good correspondence, and it found that deep fluid has an important control action on the regional seismicity.

Together, we speculated that the meteoric water, firstly, infiltrated underground and got heated by heat sources. Based on our findings, we reinforce that the relationship between geothermal fluids and seismic activity can provide insight into a better search for earthquake precursors.

**Supplementary Materials:** The following supporting information can be downloaded at: https://www.mdpi.com/article/10.3390/w14162525/s1, Table S1: The statistical table of main quantity elements of hot spring water samples; Table S2: Concentrations of trace elements in the water samples from the 20 sites; Table S3: Z score normalization of ion concentrations were calculated for comparing with the geochemical background of different springs.

**Author Contributions:** Conceptualization, Z.L. (Zirui Li) and X.Z. (Xiaocheng Zhou); methodology, Q.X. (Qiulong Xu) and Y.Y. (Yucong Yan); software, Y.L. (Ying Li) and M.H. (Miao He); validation, F.D. (Fenghe Ding) and J.T. (Jiao Tian); formal analysis, X.W.; investigation, X.W. (Xiaotao Wang); resources, C.M. (Chongzhi Ma); data curation, J.D. (Jinyuan Dong); writing—original draft preparation, Z.L. (Zhixin Luo); writing—review and editing, J.L. (Jingchao Li); visualization, X.Z. (Xiaocheng Zhou). All authors have read and agreed to the published version of the manuscript.

**Funding:** The work was funded by Central Public-Interest Scientific Institution Basal Research Fund (CEAIEF20220507, CEAIEF2022030205, CEAIEF20220213, CEAIEF2022030200, 2021IEF0101, 2021IEF1201), National Key Research and Development Project (2017YFC1500501-05, 2019YFC1509203, 2018YFE0109700) and the National Natural Science Foundation of China (41673106, 42073063, 4193000170, U2039207), IGCP Project 724 and Natural Science Foundation Project of Ningxia Hui Autonomous Region, China (2021AAC03485).

**Data Availability Statement:** Not applicable.

**Acknowledgments:** The authors would like to express their gratitude for the expert linguistic services provided accessed on 28 June 2022.

**Conflicts of Interest:** The authors declare no conflict of interest.

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
