# Peer review of "Hydrochemical Characteristics of Hot Springs in the Intersection of the Red River Fault Zone and the Xiaojiang Fault Zone, Southwest Tibet Plateau"

_water, doi:10.3390/w14162525_

Round 1
Reviewer 1 Report
The re-submitted paper presented some improvements in terms of quality and readability. However, there are still several issues to be revised in my opinion before considering it for publication.
The more critical drawback in my opinion is the presentation of results. It is still too lengthy and fragmented. The authors should follow a logical flow. As it is, the results of the paper seem more a technical report. Authors should find the main trends from the geochemical data and interpret them focusing on the main aim and possible outlooks of their paper. This section as it is, is still too long and confounding in my opinion. I suggest to improve data analysis and put more effort in the observation of specific trends (e.g., through multivariate analysis, see e.g. Binda et al. 2020 https://doi.org/10.3390/min10121058 or through spatial observation of the chemical changes ). The authors should focus more on the results presented in sections 4.4 and 4.5 and validate their conceptual model through the presented chemical data.
Moreover, there is still the need to stress the novelty and the main aims of this paper, as well as the perspective outlooks of this research. I surely suggest to add a paragraph in the introduction considering the previous knowledge in the zone (e.g., Zhou et al. 2022 https://doi.org/10.3390/w14010132 ) and explaining why this paper gives novel insights in the area.
I moreover suggest again to describe statistical analyses tools in methods, completely overlooked, instead of mention them fragmentedly in the results (see e.g. in lines 287-292 and in lines 378-385).
Finally, authors should carefully analyze data in function of sampling frequency. Some samples were collected in different periods of the year and other springs were collected once. Authors should state this issue in the manuscript text, since seasonal trends can affect chemical results (see e.g., Chaudhuri et al. 2018 https://doi.org/10.1007/s12517-018-3665-5 ).
Reviewer 2 Report
The purposes of the manuscript water-1820020 are to investigate the hydrogeochemical characteristics of 20 hot springs from the years 2015 to 2019, including the flow path inside the fault, the change of heat storage, the source of significant substances, the water-rock reaction, and the connection with earthquakes in the intersection of the Red River Fault Zone and the Xiaojiang Fault Zone, (Southwest Tibet Plateau).
The paper appears well-structured; however, some sections must be improved. Therefore, I believe
the manuscript should be published only after major revision.
Comments (R = row#):
R=30: point 3, partially repeats point 2
R=60: after the references insert the point and not the comma
R=81: the letter c) is missing in the caption of figure 1
R=84: table 1 does not refer to the mains quantity elements, but only shows the coordinates and geology!
R=150: no information is reported on the analysis of cations and anions!
R=173: there is no information on the measurement of the redox potential (Eh).
R=188: indicate in the caption of figure 2 what the two lines represent
R=230: it would be better to use box plots to show these variations
R=257: The authors should better discuss the origin of the sulfate, it may be useful to consider the dissolution of the sulphides in cristalline rocks. Consult this paper to better understand :
Fuoco, I., et al. "Arsenic polluted waters: Application of geochemical modelling as a tool to understand the release and fate of the pollutant in crystalline aquifers." Journal of Environmental Management 301 (2022): 113796.
R=271: in reaction 4) bicarbonate is missing from the reaction products
R=278: To evaluate the chemical composition of the water it is not enough to use the Piper diagram because it does not take into account (as proposed by the authors) salinity, I suggest using a TIS salinity diagram, as proposed by:
Apollaro, C.; Vespasiano, G.; De Rosa, R.: Marini L. Use of mean residence time and flowrate of thermal waters to evaluate the volume of reservoir water contributing to the natural discharge and the related geothermal reservoir volume. Application to Northern Thailand hot springs. Geothermics 2015, 58, 62-74
R=289: why is this rock used for normalization?
R=357: what mean original software?
R= 406: indicate a bibliographic reference for the geothermal gradient Tgrad
R= 410: Which database of Phreeqc was used?
Furthermore, It could be useful to improve discussion with further diagrams such as Cl vs δ18O and Cl vs. δD or others mobile costituent. Mobile constituents help comprehension about relationships and evolutions of systems. In this case, to improve knowledge about mixing between different sources, as proposed by:
Vespasiano, Giovanni, Luigi Marini, Francesco Muto, Luis F. Auqué, Mara Cipriani, Rosanna De Rosa, Salvatore Critelli et al. "Chemical, isotopic and geotectonic relations of the warm and cold waters of the Cotronei (Ponte Coniglio), Bruciarello and Repole thermal areas,(Calabria-Southern Italy)." Geothermics 96 (2021): 102228.
Cioni, Roberto, and Luigi Marini. A thermodynamic approach to water geothermometry. Springer, 2020.
R=573 The conclusions should be revised taking into account the previous comments
ADD THESE REFERENCES
Apollaro, C.; Vespasiano, G.; De Rosa, R.: Marini L. Use of mean residence time and flowrate of thermal waters to evaluate the volume of reservoir water contributing to the natural discharge and the related geothermal reservoir volume. Application to Northern Thailand hot springs. Geothermics 2015, 58, 62-74
Cioni, Roberto, and Luigi Marini. A thermodynamic approach to water geothermometry. Springer, 2020.
Vespasiano, Giovanni, Luigi Marini, Francesco Muto, Luis F. Auqué, Mara Cipriani, Rosanna De Rosa, Salvatore Critelli et al. "Chemical, isotopic and geotectonic relations of the warm and cold waters of the Cotronei (Ponte Coniglio), Bruciarello and Repole thermal areas,(Calabria-Southern Italy)." Geothermics 96 (2021): 102228.
Fuoco, I., et al. "Arsenic polluted waters: Application of geochemical modelling as a tool to understand the release and fate of the pollutant in crystalline aquifers." Journal of Environmental Management 301 (2022): 113796.
Author Response
Dear review teacher:
Thank you very much for taking time out of your busy schedule to review my article and give your very pertinent suggestions and opinions.Here are my responses to each of your requests.
Comments (R = row#):
(1)R=30: point 3, partially repeats point 2
Responses: Duplicate parts have been deleted("The circulation depths for springs were estimated in the range of 1.1 to 7.2 km.") R=29
(2)R=60: after the references insert the point and not the comma
Responses:Has been modified. R=59
(3)R=81: the letter c) is missing in the caption of figure 1
Responses:The letter c) is added to the caption of figure 1. R=85-86
(4)R=84: table 1 does not refer to the mains quantity elements, but only shows the coordinates and geology!
Responses:The name of Table 1 has been changed R=87
(5)R=150: no information is reported on the analysis of cations and anions!
Responses:Added the analysis of cations and anions. R=152-161
(6)R=173: there is no information on the measurement of the redox potential (Eh).
Responses:The redox potential had not be measured.
(7)R=188: indicate in the caption of figure 2 what the two lines represent
Responses:Added represention to the caption of figure 2. R=220-221
(8)R=230: it would be better to use box plots to show these variations
Responses:Added, see Figure 4. R=262-272
(9)R=257: The authors should better discuss the origin of the sulfate, it may be useful to consider the dissolution of the sulphides in cristalline rocks. Consult this paper to better understand :
Responses:In this part, I have read the literature you recommended, and finally I keep the original description.
Fuoco, I., et al. "Arsenic polluted waters: Application of geochemical modelling as a tool to understand the release and fate of the pollutant in crystalline aquifers." Journal of Environmental Management 301 (2022): 113796.没åŠ
(10)R=271: in reaction 4) bicarbonate is missing from the reaction products
Responses:Has been changed . R=318
(11)R=278: To evaluate the chemical composition of the water it is not enough to use the Piper diagram because it does not take into account (as proposed by the authors) salinity, I suggest using a TIS salinity diagram, as proposed by:
Responses:TIS diagram and corresponding analysis have been added. R=279-330
Apollaro, C.; Vespasiano, G.; De Rosa, R.: Marini L. Use of mean residence time and flowrate of thermal waters to evaluate the volume of reservoir water contributing to the natural discharge and the related geothermal reservoir volume. Application to Northern Thailand hot springs. Geothermics 2015, 58, 62-74
(12)R=289: why is this rock used for normalization?
Responses:Added description. R=345-348
(13)R=357: what mean original software?
Responses:Has been corrected:origin software. R=406.
(14)R= 406: indicate a bibliographic reference for the geothermal gradient Tgrad
Responses:Has been added. R=483.
(15)R= 410: Which database of Phreeqc was used?
Responses:Use the phreeqc.dat database,which was the software default.
(16)Furthermore, It could be useful to improve discussion with further diagrams such as Cl vs δ18O and Cl vs. δD or others mobile costituent. Mobile constituents help comprehension about relationships and evolutions of systems. In this case, to improve knowledge about mixing between different sources, as proposed by:
Vespasiano, Giovanni, Luigi Marini, Francesco Muto, Luis F. Auqué, Mara Cipriani, Rosanna De Rosa, Salvatore Critelli et al. "Chemical, isotopic and geotectonic relations of the warm and cold waters of the Cotronei (Ponte Coniglio), Bruciarello and Repole thermal areas,(Calabria-Southern Italy)." Geothermics 96 (2021): 102228.
Cioni, Roberto, and Luigi Marini. A thermodynamic approach to water geothermometry. Springer, 2020.
Responses:Added, see Figure 10 R=397-417
(17)R=573 The conclusions should be revised taking into account the previous comments
Responses:This paper has modified some data processing methods, but the conclusion remains the same.
(18)ADD THESE REFERENCES
Responses: Added 1-3. R=774,795,802
[1]Apollaro, C.; Vespasiano, G.; De Rosa, R.: Marini L. Use of mean residence time and flowrate of thermal waters to evaluate the volume of reservoir water contributing to the natural discharge and the related geothermal reservoir volume. Application to Northern Thailand hot springs. Geothermics 2015, 58, 62-74
[2]Cioni, Roberto, and Luigi Marini. A thermodynamic approach to water geothermometry. Springer, 2020.
[3]Vespasiano, Giovanni, Luigi Marini, Francesco Muto, Luis F. Auqué, Mara Cipriani, Rosanna De Rosa, Salvatore Critelli et al. "Chemical, isotopic and geotectonic relations of the warm and cold waters of the Cotronei (Ponte Coniglio), Bruciarello and Repole thermal areas,(Calabria-Southern Italy)." Geothermics 96 (2021): 102228.
[4]Fuoco, I., et al. "Arsenic polluted waters: Application of geochemical modelling as a tool to understand the release and fate of the pollutant in crystalline aquifers." Journal of Environmental Management 301 (2022): 113796.
Supplementary Note: In this revision, the literature recommended by the review teacher has been added again, and some irrelevant articles have been deleted.

Round 2
Reviewer 1 Report
The revised version of this paper evidently improved in terms of data interpretation and presentation, which make the paper suitable for publication in my opinion. There are, still, some minor issues to be revised in my opinion:
-line 181: "data" instead of "date";
-line 182: "Five sampling campagins were performed";
-line 187: "So" instead of "SO";
-line 514-516: please rephrase this sentence, it is unclear to me;
-line 557: "It was indicated.." instead of "It was indicates..".
Reviewer 2 Report
ok
Author Response
Please see the attachment.

This manuscript is a resubmission of an earlier submission. The following is a list of the peer review reports and author responses from that submission.
Round 1
Reviewer 1 Report
This paper present various hydrogeochemical data related with thermal springs in a seismically active area in China. While the paper reports a general robust dataset, there are different hindrances for a clear comprehension of the proper aim of the study, the paper is extremely long, discussion is often unclear and the novelty of this paper is actually limited.
First, the paper is extremely long, with different unnecessary text. Different sections related with general groundwater chemistry composition and mineral dissolution does not worth discussion in this type of paper (see e.g. in lines 106-123, 245-282 and 310-322. Similarly, there are too many graphs and figures, which often are redundant and make not clear which are the trends the authors want to highlight. For instance, hydrogen and oxygen isotopic values are added both in table 4 and figure 2, and Sr isotopes are added both in table 4 and figure 5. Then, table 8 and figure 7 are redundant.
It is also not clear from the text if the data presented in all the graphs the average along the whole time windows of every spring.
Another major drawback is the limited novelty of the paper. There is a similar data analysis to another study in a similar geological and hydrogeological setting (Zhou et al. 2022 https://doi.org/10.3390/w14010132 ). I suggest to improve data analysis and put more effort in the observation of specific trends (e.g., see e.g. Binda et al. 2020 https://doi.org/10.3390/min10121058 or through spatial observation of the chemical changes ) and to put more focus on the more fascinating results obtained in section 5.4.
Another major hindrance of the paper is its structure. The aims of the study are not well indicated in the introduction, which is vague and present different unclear sentences (lines 47-50 and 63-65, for instance).
Then, in the method section the statistical techniques used to analyze the chemical data are completely overlooked, and finally some of them are fragmentedly discussed in the results (see e.g. in lines 288-293, 371-375 and 446-464).
I finally suggest a general revision regarding the language use, since there are several misspellings and grammar errors (e.g., implaying instead of implying, PH instead of pH, anaverage instead of an average in lines 41, 170 and 180 respectively.